# Host and viral determinants for efficient SARS-CoV-2 infection of the human lung

Hin Chu [1,2,8], Bingjie Hu[2,8], Xiner Huang [2,8], Yue Chai[2,8], Dongyan Zhou[2], Yixin Wang[2], Huiping Shuai[2], Dong Yang[2], Yuxin Hou[2], Xi Zhang[2], Terrence Tsz-Tai Yuen[2], Jian-Piao Cai[2], Anna Jinxia Zhang[1,2], Jie Zhou [1,2], Shuofeng Yuan[1,2], Kelvin Kai-Wang To [1,2,3,4], Ivy Hau-Yee Chan[5], Ko-Yung Sit[5], Dominic Chi-Chung Foo [5], Ian Yu-Hong Wong[5], Ada Tsui-Lin Ng[5], Tan To Cheung[5], Simon Ying-Kit Law [5], Wing-Kuk Au[5], Melinda A. Brindley [6], Zhiwei Chen [1,2,3], Kin-Hang Kok [1,2,3], Jasper Fuk-Woo Chan [1,2,3,4,7✉] & Kwok-Yung Yuen [1,2,3,4,7✉]

Understanding the factors that contribute to efficient SARS-CoV-2 infection of human cells may provide insights on SARS-CoV-2 transmissibility and pathogenesis, and reveal targets of intervention. Here, we analyze host and viral determinants essential for efficient SARS-CoV-2 infection in both human lung epithelial cells and ex vivo human lung tissues. We identify heparan sulfate as an important attachment factor for SARS-CoV-2 infection. Next, we show that sialic acids present on ACE2 prevent efficient spike/ACE2-interaction. While SARS-CoV infection is substantially limited by the sialic acid-mediated restriction in both human lung epithelial cells and ex vivo human lung tissues, infection by SARS-CoV-2 is limited to a lesser extent. We further demonstrate that the furin-like cleavage site in SARS-CoV-2 spike is required for efficient virus replication in human lung but not intestinal tissues. These findings provide insights on the efficient SARS-CoV-2 infection of human lungs.

[1] State Key Laboratory of Emerging Infectious Diseases, Li Ka Shing Faculty of Medicine, The University of Hong Kong, Pokfulam, Hong Kong SAR, China. [2] Department of Microbiology, Li Ka Shing Faculty of Medicine, The University of Hong Kong, Pokfulam, Hong Kong SAR, China. [3] Carol Yu Centre for Infection, Li Ka Shing Faculty of Medicine, The University of Hong Kong, Pokfulam, Hong Kong SAR, China. [4] Department of Microbiology, Queen Mary Hospital, Pokfulam, Pokfulam, Hong Kong SAR, China. [5] Department of Surgery, Li Ka Shing Faculty of Medicine, The University of Hong Kong, Pokfulam, Hong Kong SAR, China. [6] Department of Infectious Diseases, Department of Population Health, College of Veterinary Medicine, University of Georgia, Athens, GA 30602, USA. [7] Hainan Medical University-The University of Hong Kong Joint Laboratory of Tropical Infectious Diseases, The University of Hong Kong, Pokfulam, Hong Kong SAR, China. [8] These authors contributed equally: Hin Chu, Bingjie Hu, Xiner Huang, Yue Chai. ✉email: jfwchan@hku.hk; kyyuen@hku.hk

Severe acute respiratory syndrome coronavirus 2 (SARS-CoV-2) was first reported in December 2019[1]. Infection by SARS-CoV-2 results in Coronavirus Disease 2019 (COVID-19), which shares a number of similar clinical manifestations with that of SARS but with a lower mortality rate[2–4]. Although less pathogenic, SARS-CoV-2 is highly efficient in person-to-person transmission and has resulted in over 48 million cases with >1.2 million deaths in 218 countries or regions within ~10 months[5,6]. Investigation of the host and viral determinants that contribute to efficient SARS-CoV-2 infection of human cells is a key research question that may facilitate our understanding of the biology of SARS-CoV-2 transmission and pathogenesis.

Successful coronavirus infection requires direct interaction between the coronavirus spike protein and the host cell surface receptor[7,8]. In addition to their protein receptors, coronavirus spikes recognize a broad array of cell surface molecules, which serve to facilitate virus attachment or entry[9–15]. Similar to SARS-CoV and human coronavirus NL63 (HCoV-NL63), the SARS-CoV-2 spike protein recognizes angiotensin-converting enzyme 2 (ACE2) as the cell receptor for entry[16,17]. Surprisingly, the expression level of ACE2 in the lung and respiratory tract is relatively low in comparison to many other human tissues, suggesting that low levels of ACE2 are biologically relevant in supporting SARS-CoV-2 entry[18]. Alternatively, SARS-CoV-2 may be able to utilize other host determinants along the respiratory tract to facilitate virus infection. In this regard, heparan sulfate (HS) and sialic acids are two ubiquitously expressed host factors that are frequently utilized by human and animal coronaviruses for attachment and entry[11–15]. However, their involvement in SARS-CoV-2 infection has not been explored.

In addition to recognition of cell surface molecules by the spike protein for attachment and entry, proteolytic activation of the coronavirus spike protein at the $S_1/S_2$ and $S_2'$ sites is essential for membrane fusion and infection of host cells[8]. Cleavage of spike begins during virus egress along the secretory pathways by proprotein convertases such as furin[19]. Additional spike cleavage occurs during virus entry and is mediated by host proteases including transmembrane protease serine 2 (TMPRSS2) and cathepsin L, which are the representative protease for the plasma membrane entry and endosomal entry pathway, respectively[20–22]. Recent studies identified a four amino acid insertion at the $S_1/S_2$ junction of SARS-CoV-2 spike that resulted in an "RRAR" sequence, which corresponded to a canonical furin-like cleavage site[23,24]. These residues are absent in the spike protein of SARS-CoV, SARS-CoV-related coronavirus (SARSr-CoV), and the closely related bat coronavirus RaTG13[25]. The additional furin-like cleavage site is widely postulated to facilitate virus entry and increase cell tropism, leading to efficient virus spread in the human population[23–26]. However, the physiological relevance of this furin-like motif in SARS-CoV-2 spike has not been functionally evaluated during infectious virus infection.

In this study, using human lung epithelial cells and ex vivo human lung tissue explants, we investigated the involvement of cell surface HS and sialic acids in SARS-CoV-2 infection and evaluated the physiological importance of the furin-like cleavage site in SARS-CoV-2 spike during virus replication. Our results reveal important information on the host and viral determinants that orchestrate the efficient SARS-CoV-2 infection of the human lungs. These findings contribute to our understanding of the biology of the high SARS-CoV-2 transmissibility and suggest targets of intervention against COVID-19.

## Results

### HS serves as an essential host determinant during SARS-CoV-2 attachment and replication.
ACE2 was identified as the receptor of SARS-CoV-2. Interestingly, ACE2 expression is relatively low in the respiratory tract and SARS-CoV-2 infection could be detected in cells with limited ACE2 expression, suggesting the involvement of additional host factors in SARS-CoV-2 infection[27,28]. HS proteoglycans were previously identified as the attachment receptors for HCoV-NL63, which served an indispensable role during HCoV-NL63 attachment and infection of target cells[11,29]. Despite both HCoV-NL63 and SARS-CoV-2 utilize ACE2 as the entry receptor, the involvement of HS proteoglycans during SARS-CoV-2 entry remain unexplored. To this end, we evaluated the role of HS during SARS-CoV-2 infection. We infected Calu3 (lung epithelial) and Caco2 (intestine epithelial) cells with early passage SARS-CoV-2 that were pretreated with HS, and harvested samples at 24 hours post infection. Our results showed that the HS treatment significantly reduced SARS-CoV-2 replication in both cell lysate and supernatant samples of Calu3 and Caco2 (Fig. 1a,b). Importantly, we further demonstrated that the HS treatment reduced SARS-CoV-2 replication in a dose-dependent manner. At 500 μg/ml, the treatment substantially reduced the amount of SARS-CoV-2 in the supernatant of Calu3 and Caco2 cells by 96% ($p < 0.0001$) and 99% ($p < 0.0001$), respectively (Fig. 1a, b). Next, to explore the mechanism of exogenous HS-mediated SARS-CoV-2 inhibition, we pretreated SARS-CoV-2 with HS for 1 hour before adding the inoculum to Calu3 and Caco2 cells at 4° for 2 hours (Fig. 1c). Our data demonstrated that exogenous HS pretreatment significantly reduced SARS-CoV-2 attachment on Calu3 and Caco2 cells by 48% ($p = 0.0032$) and 55% ($p = 0.0136$), respectively (Fig. 1d). To further confirm the role of HS during SARS-CoV-2 replication, we treated Calu3 and Caco2 cells with heparinases, which contained endoglycosidase activity that cleaved and degraded the polymeric HS molecules at the cell surface. In line with the HS treatment results, heparinase treatment efficiently reduced SARS-CoV-2 replication in both Calu3 and Caco2 cells (Fig. 1e, f). Specifically, the heparinase I + III treatment reduced SARS-CoV-2 genome copy retrieved from the supernatant of Calu3 and Caco2 cells by 73% ($p = 0.0037$) and 81% ($p = 0.0009$), respectively. Next, to evaluate the physiological relevance of HS during SARS-CoV-2 infection, we challenged ex vivo human lung tissue explants with HS- or mock-treated SARS-CoV-2 (Fig. 1g). Our result demonstrated that the HS treatment significantly reduced SARS-CoV-2 replication in the tissue homogenate of human lung tissues by 39% ($p = 0.0164$) (Fig. 1h). In addition to SARS-CoV-2, intervening the interaction between cell surface HS and virus particles similarly reduced SARS-CoV attachment and replication (Fig. 1a–h), suggesting HS played a conserved role during SARS-CoV, SARS-CoV-2, and HCoV-NL63 infection, which all utilized ACE2 for entry. Collectively, our results identify HS as an essential attachment receptor for SARS-CoV-2 infection that plays an important physiological role for efficient SARS-CoV-2 replication in the human lungs.

### Differential role of sialic acids during the attachment and replication of SARS-CoV-2, SARS-CoV, and MERS-CoV.
Sialic acid was previously identified to serve as an attachment factor for MERS-CoV[12]. We next investigated if sialic acid could similarly facilitate SARS-CoV-2 attachment and replication. We treated Calu3 and Caco2 cells with neuraminidase (NA) from *Arthrobacter ureafaciens* for 1 hour to remove cell surface sialic acids, followed by virus infection, and quantified virus production in the cell lysates and supernatants at 24 hours post infection. We observed dramatic differences on the outcome of NA treatment in SARS-CoV-2, SARS-CoV, and MERS-CoV replication. In the lung epithelial Calu3 cells, NA treatment significantly reduced MERS-CoV replication as evidenced by

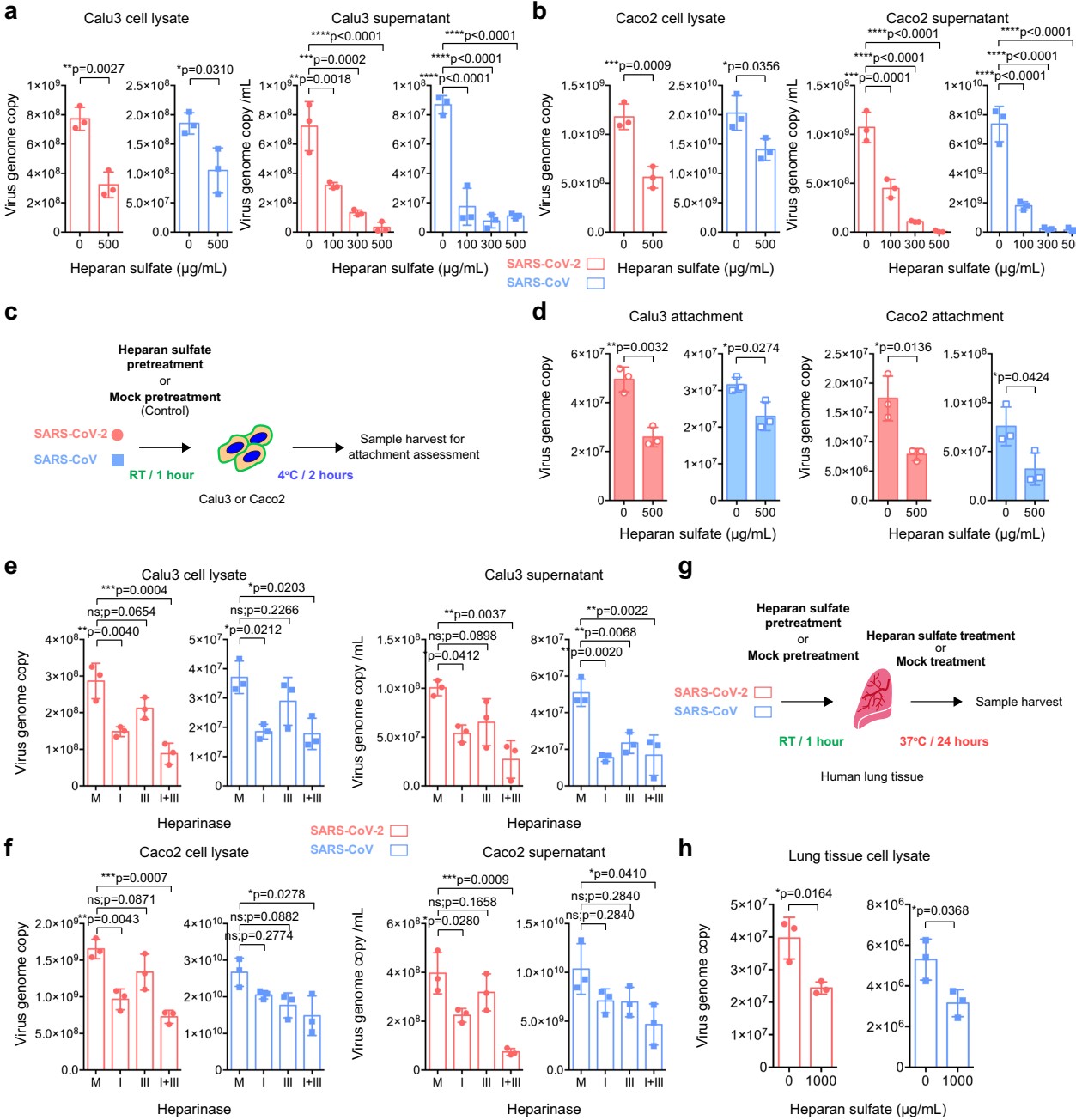

**Fig. 1 Heparan sulfate (HS) serves as an attachment factor for SARS-CoV-2 infection. a–b** Calu3 and Caco2 cells were inoculated with HS-pretreated SARS-CoV-2 or SARS-CoV at 0.2 MOI for 2 h at 37 °C. Cell lysates and supernatants were harvested at 24hpi and virus replication was determined with qRT-PCR ($n = 3$). **c** Schematic of the attachment assay. **d** Calu3 and Caco2 cells were inoculated with HS-pretreated SARS-CoV-2 or SARS-CoV at 0.2 MOI for 2 h at 4 °C. Cell lysates were harvested after the 2 h incubation and virus attachment was determined with qRT-PCR ($n = 3$). **e–f** Calu3 and Caco2 were pretreated with Heparinase I (4U/ml), Heparinase III (0.5U/ml) or both for 1 h at 37 °C, followed by virus infection at 0.2 MOI. Cell lysates and supernatants were harvested at 24hpi for qRT-PCR analysis ($n = 3$). **g–h** Human lung tissues were challenged with HS-pretreated SARS-CoV-2 or SARS-CoV at an inoculum of $1 \times 10^7$ PFU/ml for 2 h. Tissues were harvested and homogenized at 24hpi for viral gene copy detection ($n = 3$). Data represented mean and standard deviations from the indicated number of biological repeats. Statistical significance between groups was determined with one way-ANOVA (**a** and **b** supernatant panels, **e** and **f**) or two-sided unpaired Student's $t$ test (**a** and **b** cell lysate panels, **d** and **h**). * represented $p < 0.05$, ** represented $p < 0.01$, *** represented $p < 0.001$, **** represented $p < 0.0001$. *ns* not significant. Source data are provided as a Source Data file.

the sharp decrease in virus genome copy in cell lysate and supernatant samples (Fig. 2a), which is in line with previous reports[12,30]. The positive role of sialic acid on MERS-CoV replication was further confirmed with plaque assays, which demonstrated an 86% ($p < 0.0001$) decrease in infectious virus titer upon NA treatment in comparison with the mock-treated samples (Fig. 2a). In stark contrast to MERS-CoV, NA

treatment did not decrease but significantly increased SARS-CoV replication. This was confirmed by both real-time quantitative PCR (qRT-PCR) and plaque assays, which showed a 492% ($p < 0.0028$) increase in infectious virus titer upon NA treatment (Fig. 2a). In this regard, our data demonstrated an opposing role for sialic acid on MERS-CoV and SARS-CoV infection, suggesting that while sialic acid facilitated MERS-

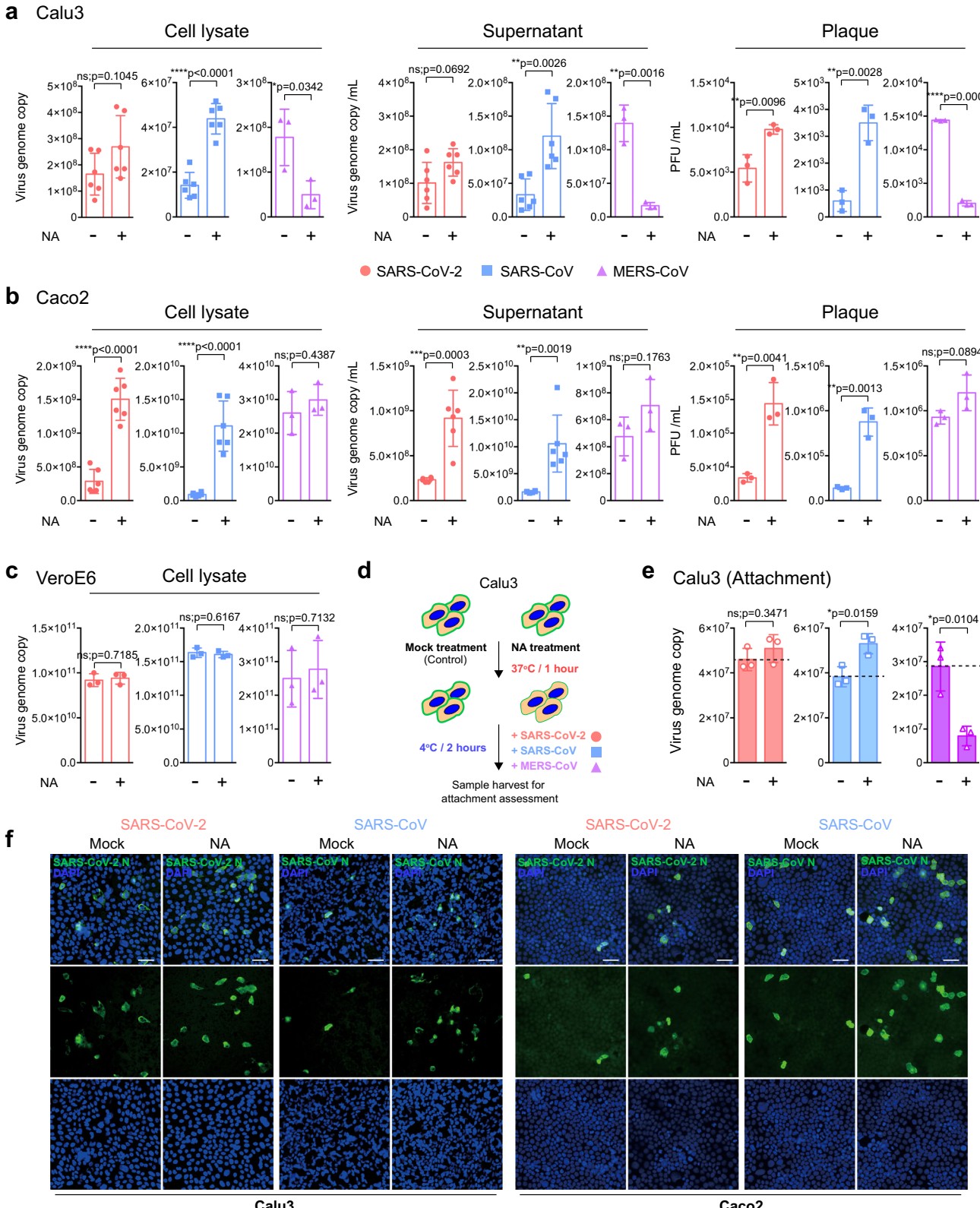

CoV infection, it served to restrict SARS-CoV infection in Calu3 cells. Next, we evaluated the consequence of NA treatment on SARS-CoV-2 infection. Interestingly, NA treatment modestly increased SARS-CoV-2 infectious particle production by 80.3% ($p = 0.0096$) (Fig. 2a), suggesting that SARS-CoV-2 could partly overcome the sialic acid-mediated restriction in Calu3 cells in comparison with that of SARS-CoV. In Caco2

cells, whereas NA treatment did not modulate MERS-CoV replication, it significantly promoted the replication of both SARS-CoV-2 and SARS-CoV (Fig. 2b). In contrast to Calu3 and Caco2 cells, NA treatment did not modulate virus replication for all three viruses in VeroE6 cells (Fig. 2c), which was potentially explained by the difference in the level of cell surface sialic acid expression as suggested from a previous report[12].

**Fig. 2 Sialic acids play differential roles on attachment and replication of SARS-CoV-2, SARS-CoV, and MERS-CoV. a–c** Calu3, Caco2, and VeroE6 cells were treated with neuraminidase (NA) for 1 h at 37 °C, followed by SARS-CoV-2, SARS-CoV, or MERS-CoV inoculation at 0.2 MOI for 2 h. Cell lysates and supernatants were harvested at 24hpi for qRT-PCR analysis and plaque assay titration ($n = 6$ for SARS-CoV-2 and SARS-CoV cell lysate or supernatant samples in **a** and **b**, $n = 3$ for other panels). **d** Schematic of attachment assay. **e** NA-pretreated Calu3 cells were incubated with the viruses at 4 °C for 2 h. Cell lysates were harvested at 2hpi for qRT-PCR analysis ($n = 3$). **f** NA- or mock-treated Calu3 or Caco2 cells were challenged with SARS-CoV-2 or SARS-CoV at 0.2 MOI. Infected cells were fixed at 16hpi and immunolabeled for virus nucleocapsid (N) protein (green) and DAPI (blue). Images were taken at ×20 magnification. Bars represented 50 µm. Data represented mean and standard deviations from the indicated number of biological repeats. The experiment in **f** was repeated three times independently with similar results. Statistical significance between groups was determined with two-sided unpaired Student's t-test. * represented $p < 0.05$, ** represented $p < 0.01$, *** represented $p < 0.001$, **** represented $p < 0.0001$. *ns* not significant. Source data are provided as a Source Data file.

To investigate if sialic acid directly impacted the attachment of SARS-CoV-2, SARS-CoV, and MERS-CoV on cell surface, we pretreated Calu3 cells with NA for 1 hour and inoculated the cells with individual viruses at 4 °C for 2 hours (Fig. 2d). Our result demonstrated that while the NA treatment reduced MERS-CoV attachment by 72% ($p = 0.0104$), it increased SARS-CoV attachment by 39% ($p = 0.0159$) and marginally increased SARS-CoV-2 attachment by 10% ($p = 0.3471$) (Fig. 2e). In addition, we further evaluated the role of sialic acids on SARS-CoV-2 and SARS-CoV infection with immunofluorescence detection of the viral nucleocapsid (N) protein. The imaging results indicated that although NA treatment modestly increased SARS-CoV-2 infection in Calu3, the increase was more evident for SARS-CoV. In Caco2 cells, NA treatment efficiently promoted the infection of both viruses (Fig. 2f).

The differential interplay with sialic acids during SARS-CoV-2 and SARS-CoV infection is potentially important and may contribute to the difference in the efficiency of infecting and replicating in human lung cells by the two viruses[31,32]. To further investigate the interaction of SARS-CoV-2 and SARS-CoV with sialic acids, we employed the solute carrier family 35 member A1 (SLC35A1) knockout HEK293 cells, which are deficient in cell surface sialic acid expression and are resistant to influenza virus infection (Fig. 3a and Supplementary Figure 1)[33]. With ACE2 overexpression (Fig. 3b), the attachment of SARS-CoV on the cell surface of the control knockout cells (SLC35A1$^{WT}$/ACE2$^{oe}$) was 868% than that of the mock-overexpressed cells (SLC35A1$^{WT}$/ACE2$^m$). In ACE2-overexpressed SLC35A1 knockout cells (SLC35A1$^{KO}$/ACE2$^{oe}$), SARS-CoV attachment significantly further increased to 1423% in comparison with the control (SLC35A1$^{WT}$/ACE2$^m$) (SARS-CoV/SLC35A1$^{KO}$/ACE2$^{oe}$ vs SARS-CoV/SLC35A1$^{WT}$/ACE2$^{oe}$: $p < 0.0001$) (Fig. 3c). However, the attachment of SARS-CoV-2 was only modestly increased in the surface sialic acid deficient cells (SLC35A1$^{KO}$/ACE2$^{oe}$: 846%) in comparison with the control (SLC35A1$^{WT}$/ACE2$^{oe}$: 723%) (SARS-CoV-2/SLC35A1$^{KO}$/ACE2$^{oe}$ vs SARS-CoV-2/SLC35A1$^{WT}$/ACE2$^{oe}$: $p = 0.3052$) (Fig. 3c).

To investigate whether sialic acids played physiologically important roles during SARS-CoV-2 and SARS-CoV infection of the human lungs, we treated ex vivo human lung tissues with NA, followed by SARS-CoV-2 or SARS-CoV challenge. Our results showed that SARS-CoV-2 replicated more efficiently than SARS-CoV in the human lung without NA treatment ($p = 0.0077$ at 48 hpi) (Fig. 3d), which was consistent with our previous report[31]. The NA treatment significantly increased the replication of SARS-CoV ($p = 0.0048$ at 48 hpi) but not SARS-CoV-2 ($p = 0.1173$ at 48 hpi) in the human lung tissues, which mitigated the replication difference between SARS-CoV-2 and SARS-CoV ($p = 0.1713$ at) (Fig. 3d). The more sizable increase for SARS-CoV over SARS-CoV-2 replication upon NA treatment was additionally evidenced by the significant increase in intracellular virus genome copies and infectious virus titers in SARS-CoV but not SARS-CoV-2 infection (Fig. 3e, f). We next performed

immunohistochemistry to directly visualize the effect of NA treatment on SARS-CoV-2 and SARS-CoV infection in the human lung tissues. As demonstrated in Fig. 3g, although NA treatment dramatically increased the extent of SARS-CoV infection (middle panels, arrows) in human lung tissues, the level of SARS-CoV-2 infection was modestly affected (left panels). Taken together, our results demonstrate an unexpected role of sialic acids in restricting SARS-CoV-2 and SARS-CoV replication in human lung tissues. In comparison with SARS-CoV, infection of SARS-CoV-2 is affected to a lesser extent by sialic acid-mediated restriction, which may contribute to its more efficient infection and replication in human lungs.

**Sialic acids present on ACE2 precluded perfect spike/ACE2-interaction during SARS-CoV-2 infection.** Recent studies identified multiple glycosylation sites on ACE2[34–36]. To investigate if sialic acid present on ACE2 may have precluded perfect spike–ACE2 interaction in SARS-CoV-2 infection, we obtained ACE2 mutants that abolished glycosylation. Specifically, N53Q, N90Q, N103Q, N322Q, N432Q, N546Q, and N690Q ACE2 mutants were individually generated to obtain N-linked glycosylation ACE2 mutants (Fig. 4a, b). Two of these sites, N90 and N322, were predicted to form interactions with the spike protein[35]. Inter-molecular glycan–glycan interactions were predicted between the glycans at N546 of ACE2 and those at N74 and N165 of spike protein[35]. In addition to the N-linked glycosylation sites, O-linked glycosylation sites were identified at S155, T496, and T730[34–36]. Thus, we generated S155A, T496A, and T730A mutations to remove these O-linked glycosylation motif (Fig. 4a, b). Moreover, we generated "del N", "del O", and "del N + O" mutants that simultaneously removed all N-linked, all O-linked, and all N- and O-linked glycosylation sites, respectively (Fig. 4a, b, Supplementary Figure 2, and Supplementary Figure 3). We evaluated these ACE2 mutants on their capacity to support SARS-CoV-2 entry and SARS-CoV-2 replication. To evaluate SARS-CoV-2 entry, we produced VSV-based SARS-CoV-2-S-pseudoviruses. We transfected these ACE2 mutant constructs into BHK21 cells and transduced the cells with SARS-CoV-2-S-pseudoviruses 24 hours post transfection. Pseudovirus entry efficiency was quantified by measuring the level of luciferase signal detected at 24 hours post transduction. Our results revealed a number of important findings. First, SARS-CoV-2-S-pseudovirus entry was significantly more efficient in cells transfected with ACE2 mutants N90Q and N546Q, which increased virus entry by 60% ($p = 0.0189$) and 68% ($p = 0.006$) for N90Q and N546Q, respectively (Fig. 4c). Second, none of the single mutations significantly reduced SARS-CoV-2-S-pseudovirus entry. This suggested that individually removing any of the predicted glycosylation sites did not significantly reduce SARS-CoV-2 entry. Third, while removing all N-linked (del N) or O-linked (del O) glycosylation sites did not impact SARS-CoV-2-S-pseudovirus entry, simultaneously removing all N- and all O-linked

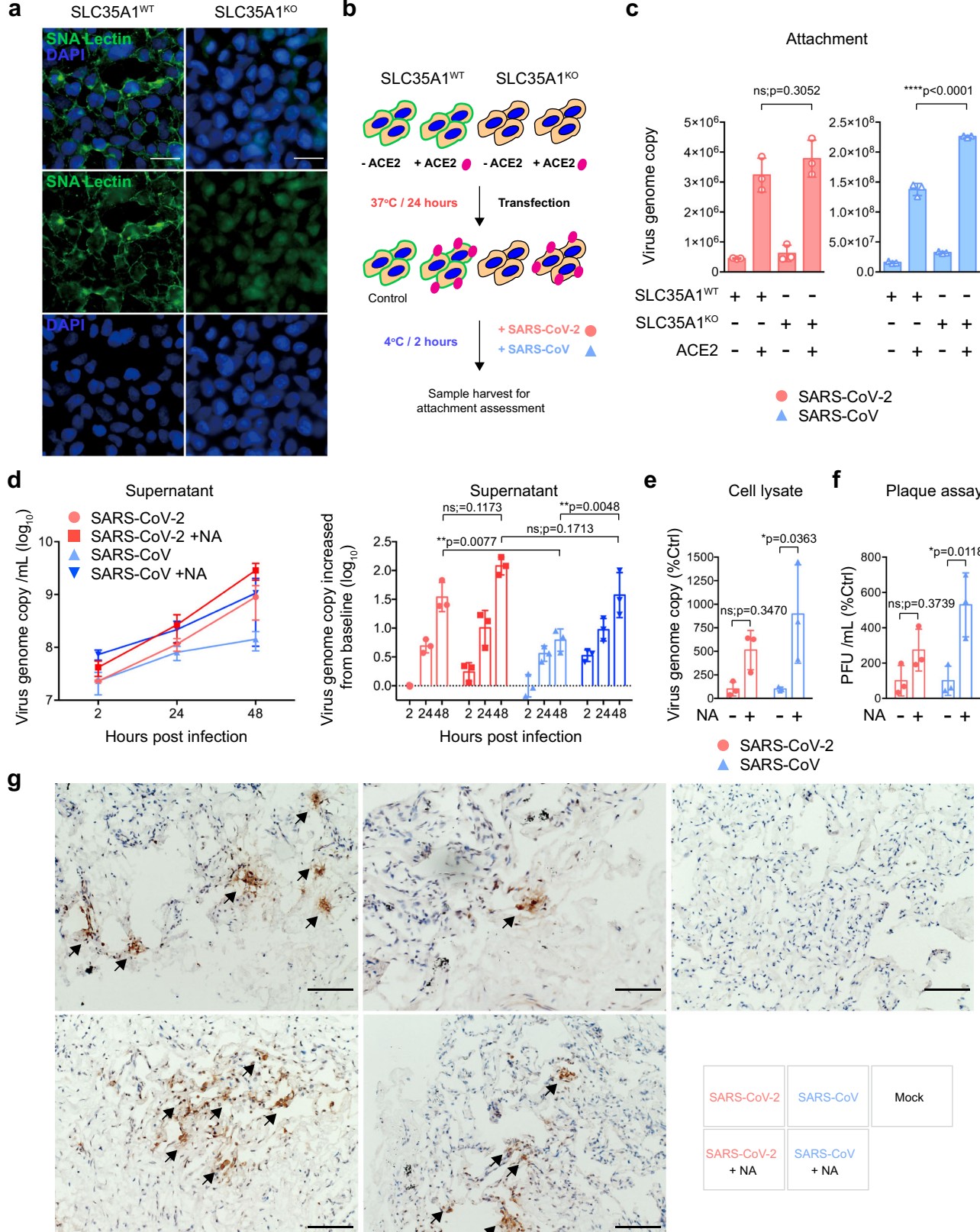

glycosylation sites dramatically reduced SARS-CoV-2-S-pseudo-virus entry ($p < 0.0001$), which might partly due to the low cell surface expression of the del N + O ACE2 mutant (Fig. 4c and Fig. Supplementary Figure 2). In contrast to the SARS-CoV-2-S-pseudovirus, the entry of VSV-G-pseudovirus was not affected by the transfection of any ACE2 mutants (Fig. 4d). Next, to evaluate

SARS-CoV-2 replication, we transfected the ACE2 mutant constructs into BHK21 cells and inoculated the cells with SARS-CoV-2. Samples were harvested at 24 hours post infection and virus replication in the cell lysates and supernatants were quantified with qRT-PCR. The results were largely in line with those obtained in the pseudovirus entry assays. In particular,

**Fig. 3 SARS-CoV-2 partly overcomes sialic acid-mediated restriction in ex vivo human lung tissues. a** SLC35A1$^{WT}$ and SLC35A1$^{KO}$ 293 cells were fixed and stained with Sambucus Nigra Lectin (green) and DAPI (blue) for cell surface sialic acid detection. Bars represented 50 μm. **b** Schematic of attachment assay. **c** SLC35A1$^{WT}$ and SLC35A1$^{KO}$ 293 cells with or without hACE2 overexpression were inoculated with SARS-CoV-2 or SARS-CoV at 0.2 MOI for 2 h at 4 °C. Cell lysates were harvested at 2hpi for qRT-PCR analysis ($n = 3$). **d–e** Mock- or NA-treated ex vivo human lung tissues were infected with SARS-CoV-2 or SARS-CoV at an inoculum of $1 \times 10^7$ PFU/ml. Supernatants were collected at 2, 24, and 48hpi and tissue samples were collected at 48hpi for qRT-PCR analysis ($n = 3$). **f** Supernatants at 48hpi were titrated by plaque assays ($n = 3$). **g** Representative images of human lung tissues challenged with SARS-CoV-2 or SARS-CoV with or without NA treatment. Viral N proteins were detected with anti-SARS-CoV-2-N or rabbit anti-SARS-CoV-N immune serum (arrows). Bars represented 100 μm. Data represented mean and standard deviations from the indicated number of biological repeats. The experiments in **a** and **g** were repeated three times independently with similar results. Statistical significance between groups was determined with two way-ANOVA. * represented $p < 0.05$, ** represented $p < 0.01$, *** represented $p < 0.001$, **** represented $p < 0.0001$. *ns* not significant. Source data are provided as a Source Data file.

simultaneously removing all N- and all O-linked glycosylation sites dramatically reduced SARS-CoV-2 genome copy in both cell lysate and supernatant samples (Fig. 4e, f). In this setting, the N90Q but not the N546Q mutant demonstrated a statistically significant increase over the WT control in the cell lysate samples (Fig. 4e). Overall, our experimental results using the ACE2 glycosylation mutants indicate that glycans including sialic acids at the N90 position of ACE2 have the most significant role in precluding a perfect ACE2-spike interaction during SARS-CoV-2 infection.

By introducing the N to Q or S/T to A mutations at the predicted ACE2 glycosylation sites, all glycan modifications in addition to the sialic acid groups are abolished at the modified site. To directly address the role of sialic acids present on ACE2 in spike/ACE2-interaction, we performed surface plasmon resonance (SPR) to determine the protein-protein binding affinity between SARS-CoV-2 spike receptor-binding domain (RBD) and ACE2 with or without pretreating ACE2 with neuraminidase. In this setting, human ACE2 binds to SARS-CoV-2 RBD with a equilibrium dissociation constant ($K_D$) of 37.4 nM, which is consistent with other SPR studies[37,38] (Fig. 4g). Importantly, after neuraminidase pretreatment, human ACE2 binds to SARS-CoV-2 RBD at a higher binding affinity ($K_D = 16.2$ nM) (Fig. 4h). Collectively, our findings indicate that the sialic acids present on the cellular receptor ACE2 precluded perfect spike/ACE2-interaction during SARS-CoV-2 infection.

**The inserted furin-like cleavage site in the SARS-CoV-2 spike is required for efficient virus replication in the human lung.** Recent studies identified a four amino-acid insertion at the boundary between the SARS-CoV-2 spike $S_1$ and $S_2$ subunits[23,24], resulting in a "RRAR" furin-like cleavage site in SARS-CoV-2 spike. These residues are not present at the $S_1/S_2$ cleavage site of SARS-CoV spike, SARSr-CoV spike, and the closely related bat coronavirus RaTG13 spike[25]. Similar amino-acid insertions that created a polybasic furin cleavage site were reported in the hemagglutinin protein of influenza viruses and were found to contribute to virus virulence[39]. However, the physiological importance of this furin-like cleavage site introduced at $S_1/S_2$ junction of SARS-CoV-2 spike has not been functionally evaluated during virus infection. To this end, we first overexpressed human furin and ACE2 in the non-permissive BHK21 cells (Fig. 5a), followed by SARS-CoV-2 or SARS-CoV infection, and assessed virus replication at 24 hpi. Interestingly, SARS-CoV-2 replicated to similar levels in the ACE2$^{oe}$/Furin$^{oe}$ and the ACE2$^{oe}$/Furin$^m$ BHK21 cells (Fig. 5b, c), suggesting that the expression of furin did not facilitate SARS-CoV-2 replication in the ACE2-expressing BHK21 cells. Next, we compared the replication of SARS-CoV-2 with a SARS-CoV-2 mutant (SARS-CoV-2 $S_1/S_2$mut), which contained a 10-amino acid deletion in spike that removed the $S_1/S_2$ furin-like cleavage site (Fig. 5d, Supplementary Figure 4, and Supplementary Figure 5) in Huh7

cells with or without furin overexpression (Fig. 5e). Our data demonstrated that SARS-CoV-2 $S_1/S_2$mut and SARS-CoV-2 replicated to largely comparable levels in Huh7 cells (Fig. 5f, g). At the same time, overexpression of furin in Huh7 did not promote the replication of SARS-CoV-2 $S_1/S_2$mut or SARS-CoV-2 (Fig. 5f, g).

Next, we infected a panel of seven different cell types of human and non-human origin, including Caco2 (human intestine), Calu3 (human lung), Huh7 (human liver), CRFK (cat), RK13 (rabbit), PK15 (pig), and VeroE6 (monkey) cells, with SARS-CoV-2 or SARS-CoV-2 $S_1/S_2$mut. Interestingly, SARS-CoV-2 replicated significantly better than SARS-CoV-2 $S_1/S_2$mut in Calu3 (25.2-folds; $p = 0.0046$) but not the other evaluated cells (Fig. 6a). Similarly, SARS-CoV-2-S-pseudoviruses also entered Calu3 cells more efficiently in comparison to the SARS-CoV-2-S ($S_1/S_2$mut)-pseudoviruses (4.6-folds; $p = 0.0038$) (Supplementary Figure 6). To understand why the furin cleavage site was required for efficient SARS-CoV-2 replication and entry in Calu3 but not other cells, we analyzed the expression level of representative host proteases from Caco2, Calu3, Huh7, and VeroE6 cells. Our results suggested that Caco2 cells expressed the highest level of TMPRSS2, which was 46- and 28-folds higher than that of Calu3 and Huh7, respectively. TMPRSS2 expression was under the detection limit in VeroE6 cells. At the same time, VeroE6 and Huh7 cells expressed high level of cathepsin L, which was 434-/19- and 329-/14-folds higher than that of Caco2 and Calu3 cells, respectively (Fig. 6b). Calu3 cells expressed low levels of both TMPRSS2 and cathepsin L, which were the representative enzyme responsible for plasma membrane and endosomal entry, respectively (Fig. 6b). To further evaluate the role of furin on SARS-CoV-2 replication in Calu3 cells, we examined the replication and spike cleavage of SARS-CoV-2 in Calu3 cells with or without the presence of the furin inhibitor, dec-RVKR-cmk. Our results demonstrated that dec-RVKR-cmk reduced spike cleavage and inhibited SARS-CoV-2 replication in Calu3 cells in a dose-dependent manner (Supplementary Figure 7a, b). In contrast, dec-RVKR-cmk did not inhibit spike cleavage or SARS-CoV-2 replication in VeroE6 cells (Supplementary Figure 7c and 7d), which robustly expressed cathepsin L. Overall, these findings suggested that furin cleavage at $S_1/S_2$ junction of spike was required for efficient SARS-CoV-2 replication in Calu3 cells, potentially owing to the relatively low TMPRSS2 and cathepsin L expression in Calu3 cells.

We next compared the replication of SARS-CoV-2 and SARS-CoV-2 $S_1/S_2$mut in ex vivo human lung tissues and ex vivo human intestinal tissues (Fig. 6c). Importantly, SARS-CoV-2 replicated more efficiently than that of SARS-CoV-2 $S_1/S_2$mut in the human lung tissues (Fig. 6d,e). Specifically, SARS-CoV-2 replicated 3.2-fold ($p = 0.0042$) and 2.3-fold ($p = 0.0011$) higher than that of SARS-CoV-2 $S_1/S_2$mut at 36 and 60 hpi, respectively (Fig. 6d). Interestingly, despite their differential replication capacity in human lung tissues, the two viruses replicated to

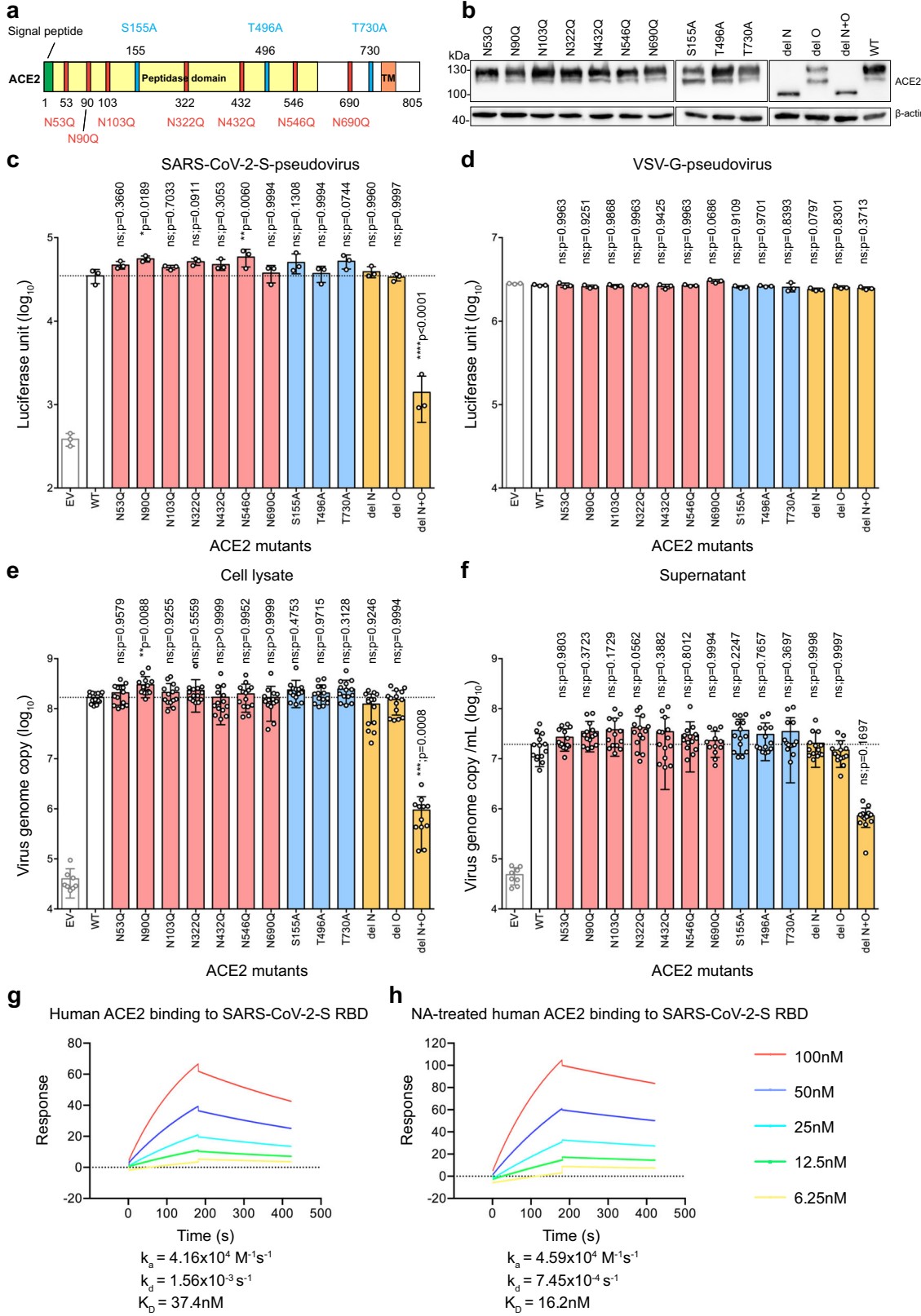

similar levels in ex vivo human intestinal tissues (Fig. 6f, g). Next, we evaluated the expression of ACE2 and representative proteases in human lung and intestinal tissues. Our results revealed that the TMPRSS2 expression level in the human intestinal tissues was 13-folds higher than that in the human lung tissues, which might explain the dependency on furin activation of the viral spike protein for SARS-CoV-2 replication in the human lung but not intestine (Fig. 6h). Overall, these results suggested that the furin-like cleavage site located at the $S_1/S_2$ junction of SARS-CoV-2 spike is required for efficient virus replication in the human lungs.

**Fig. 4 Sialic acids present on ACE2 precluded perfect spike/ACE2-interaction. a** Schematic of the N-linked and O-linked ACE2 glycosylation mutants. **b** Western blot of ACE2 glycosylation mutants overexpressed in BHK21 cells. **c** BHK21 cells were transfected with the ACE2 mutants for 24 h before inoculating the cells with SARS-CoV-2-S-pseudovirus. Pseudovirus entry was determined at 24 h post inoculation ($n = 3$). **d** Entry of VSV-G-pseudovirus in BHK21 cells transfected with the ACE2 mutants ($n = 3$). **e** BHK21 cells expressing the ACE2 mutants were infected with SARS-CoV-2. Virus replication in the cell lysate samples at 24hpi was evaluated with qRT-PCR ($n = 8$ for EV, $n = 12$ for del N + O, $n = 14$ for other groups). **f** Virus replication in the supernatant samples at 24hpi was evaluated with qRT-PCR ($n = 8$ for EV, $n = 12$ for N690Q, $n = 14$ for other groups). **g** The binding affinity between SARS-CoV-2-S RBD and human ACE2 was determined with SPR. **h** The binding affinity between SARS-CoV-2-S RBD and NA-pretreated human ACE2 was determined with SPR. Data represented mean and standard deviations from the indicated number of biological repeats. The experiments in **b**, **g**, and **h** were repeated three times independently with similar results. Statistical significance between groups was determined with one way-ANOVA. * represented $p < 0.05$ and ** represented $p < 0.01$. ns not significant, EV empty vector. Source data are provided as a Source Data file.

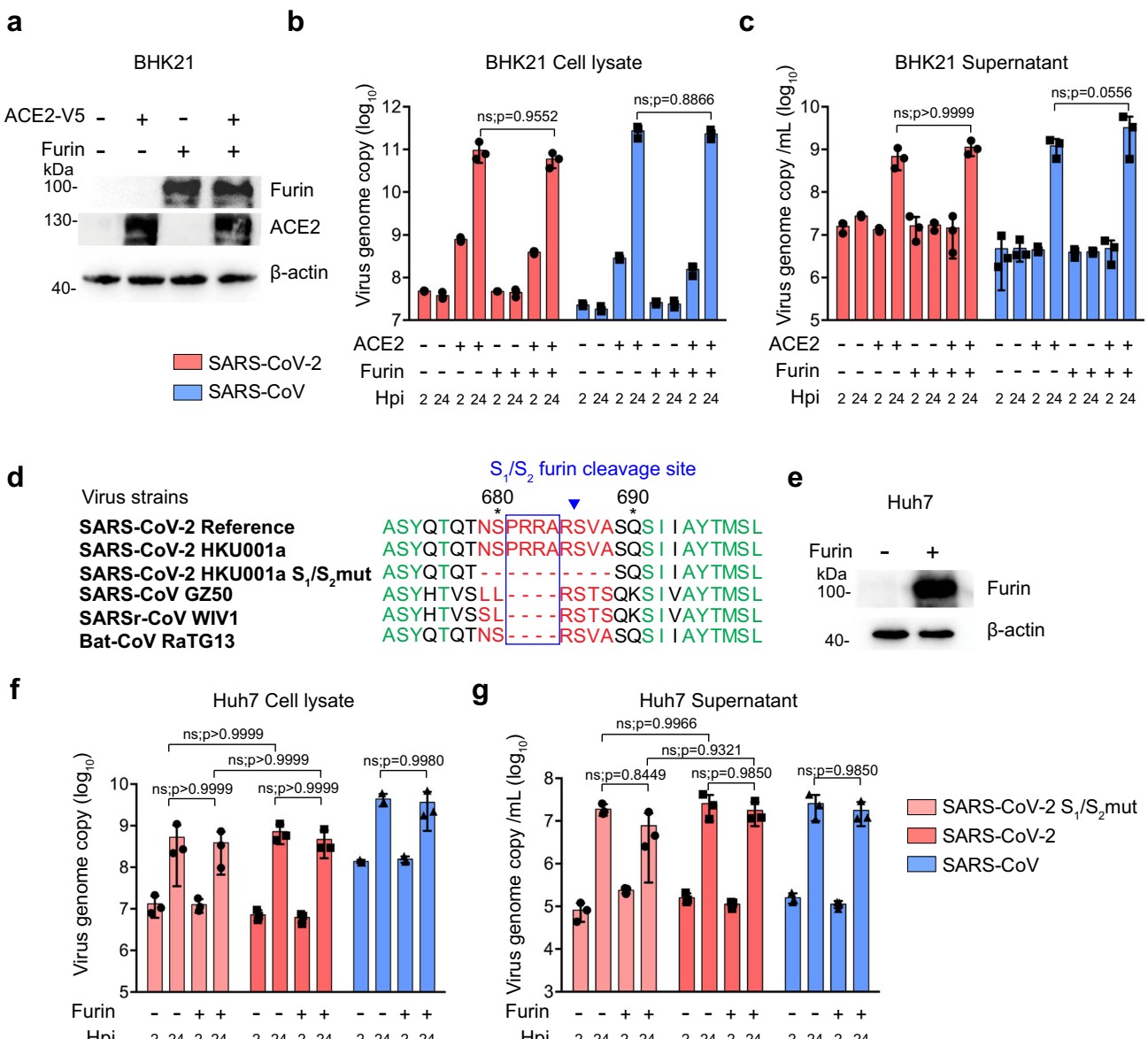

**Fig. 5 The inserted furin-like cleavage site or furin overexpression does not promote SARS-CoV-2 replication in BHK21 (non-permissive) and Huh7 (permissive) cells. a** Human ACE2 and furin overexpression in BHK21 cells was confirmed with western blot. **b–c** BHK21 cells with or without human ACE2 or furin overexpression were infected with SARS-CoV-2 or SARS-CoV. Cell lysates and supernatants were collected at 2 and 24hpi for qRT-PCR analysis ($n = 3$). **d** Amino-acid sequence alignment of residues around the $S_1/S_2$ cleavage site of SARS-CoV-2 reference strain (GenBank accession number: MN908947), SARS-CoV-2 HKU001a (MT230904), SARS-CoV-2 HKU001a $S_1/S_2$mut (MT621560), SARS-CoV GZ50 (AY304495), SARSr-CoV WIV1 (KF367457), and bat-CoV RaTG13 (MN996532). **e** Furin overexpression in Huh7 cells was confirmed with western blot. **f–g** Huh7 cells with or without furin overexpression were infected with SARS-CoV-2, SARS-CoV-2 $S_1/S_2$mut, or SARS-CoV. Cell lysates and supernatants were collected at 2 and 24hpi for qRT-PCR analysis ($n = 3$). Data represented mean and standard deviations from the indicated number of biological repeats. The experiment in **a** and **e** was repeated three times independently with similar results. Statistical significance between groups was determined with two way-ANOVA. ns not significant. Source data are provided as a Source Data file.

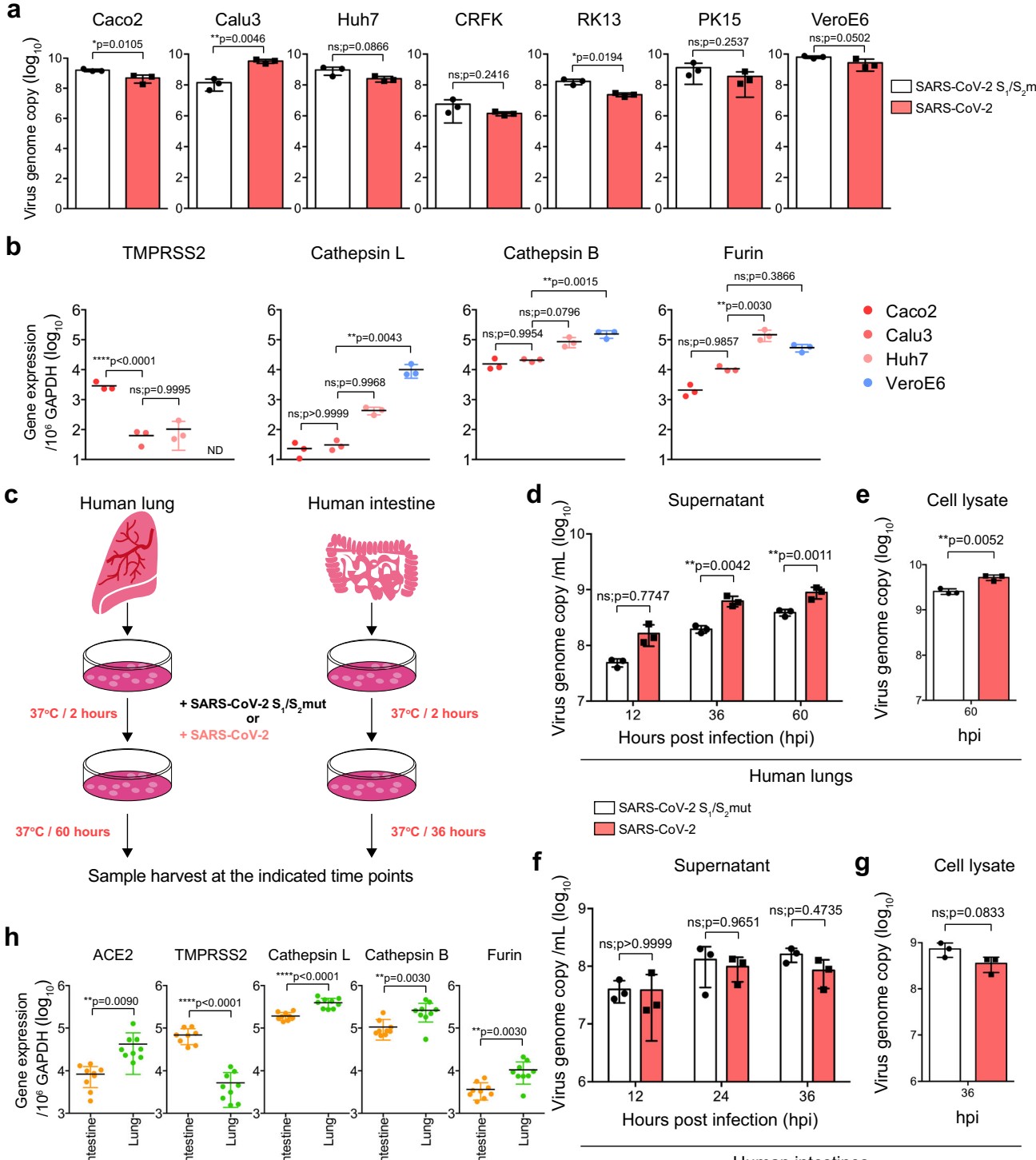

**Fig. 6 The furin-like cleavage site in SARS-CoV-2 spike is required for efficient SARS-CoV-2 replication in human lungs. a** Seven cell types of human and non-human origin, including Caco2 (human intestine), Calu3 (human lung), Huh7 (human liver), CRFK (cat), RK13 (rabbit), PK15 (pig), and VeroE6 (monkey) were infected with SARS-CoV-2 or SARS-CoV-2 $S_1/S_2$mut at 0.2 MOI. Cell lysates were collected at 24hpi for viral genome copy analysis by qRT-PCR ($n = 3$). **b** The expression of TMPRSS2, cathepsin L, cathepsin B, and furin from Caco2, Calu3, Huh7, and VeroE6 were analyzed with qRT-PCR ($n = 3$). **c** Schematic of ex vivo human lung and intestinal tissue infection. **d–g** Human lung and intestine tissues were infected with SARS-CoV-2 or SARS-CoV-2 $S_1/S_2$mut. Cell lysate and supernatant samples were harvested at the indicated time points for qRT-PCR analysis ($n = 3$). **h** The expression of ACE2, TMPRSS2, cathepsin L, cathepsin B, and furin from human lung and intestinal tissues were determined with qRT-PCR ($n = 8$ for TMPRSS2 in intestine and $n = 9$ for other groups). Data represented mean and standard deviations from the indicated number of biological repeats. Statistical significance between groups was determined with two way-ANOVA (**d** and **f**), one way-ANOVA (**b**), or two-sided unpaired Student's $t$ test (**a, e, g,** and **h**). * represented $p < 0.05$, ** represented $p < 0.01$, *** represented $p < 0.001$, **** represented $p < 0.0001$. *ns* not significant. Source data are provided as a Source Data file.

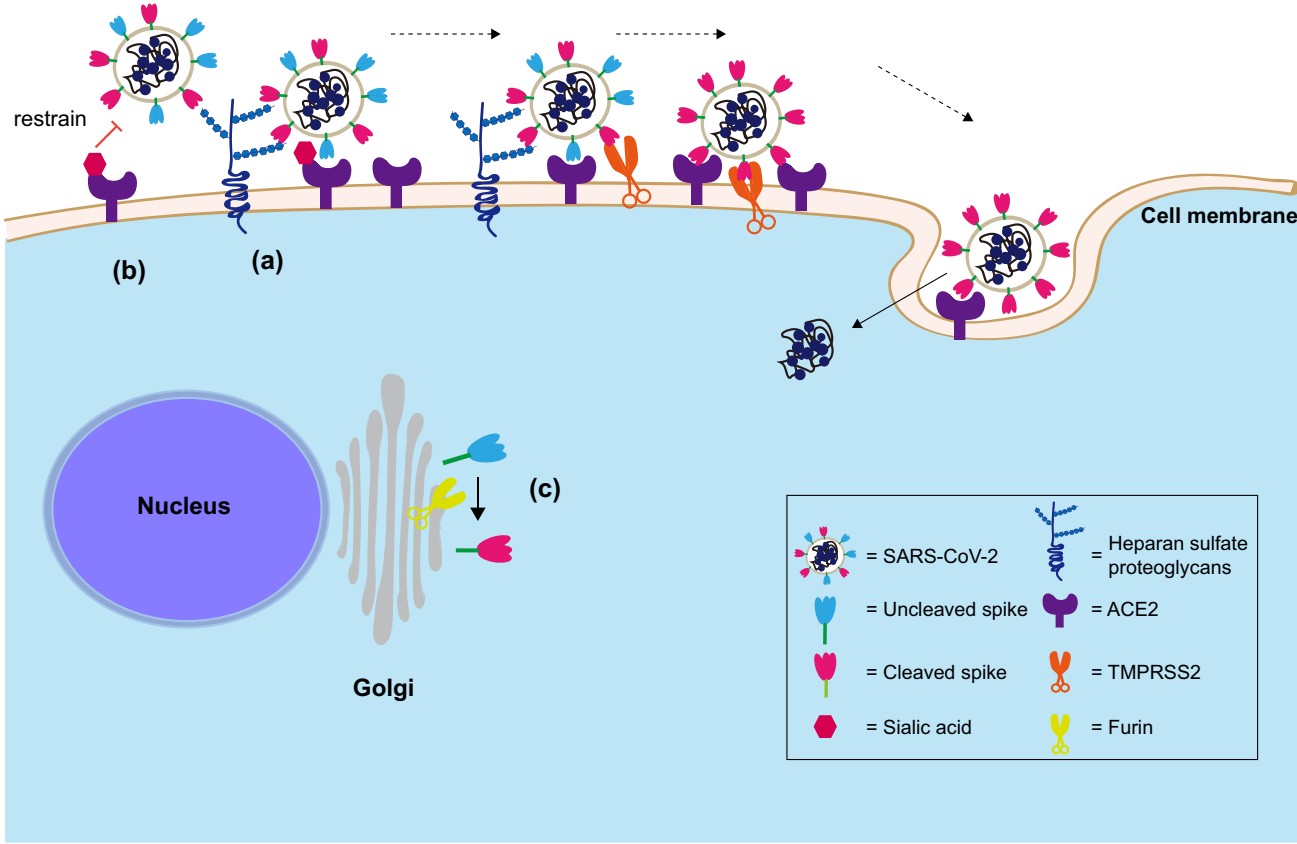

**Fig. 7 Schematic of the host and viral determinants for SARS-CoV-2 infection of the human lung as revealed in this study. a** SARS-CoV-2 can utilize cell surface heparan sulfate proteoglycan as an attachment receptor. **b** Sialic acids present on ACE2 precluded perfect spike/ACE2-interaction but SARS-CoV-2 is affected to a lesser extent by the sialic acid-mediated restriction in lung cells in comparison with SARS-CoV. **c** Furin cleavage of SARS-CoV-2 spike is essential for efficient SARS-CoV-2 replication in the human lungs, which is potentially due to the relatively low level of TMPRSS2 expression in the lungs.

## Discussion

As evidenced by the rapid dissemination of SARS-CoV-2 globally, this novel betacoronavirus is highly efficient in person-to-person transmission[5,6]. Understanding the host and viral determinants that contribute to efficient SARS-CoV-2 infection of human cells could provide insights on the biology of SARS-CoV-2 transmission and pathogenesis, and potentially reveal targets of intervention. In this study, we described key host and viral features that contributed to the efficient replication of SARS-CoV-2 in the human lungs, which were not previously reported (Fig. 7). First, we identified HS as an attachment receptor for SARS-CoV-2 infection in in vitro human lung epithelial cells and ex vivo human lung tissue explants. Second, we identified sialic acids as a host factor that restricted SARS-CoV-2 and SARS-CoV infection. More importantly, SARS-CoV-2 was capable of partly overcoming the sialic acid-mediated restriction, which facilitated its replication in the human lung tissues. Third, we identified the furin-like cleavage site at the $S_1/S_2$ junction of SARS-CoV-2 spike to be essential for efficient virus replication in human lung epithelial cells. We further showed that SARS-CoV-2 replicated to significantly higher levels in the ex vivo human lung tissues than that of SARS-CoV-2 $S_1/S_2$mut. The difference was organ-specific as the replication of the two viruses was comparable in ex vivo human intestinal tissues. By analyzing the expression of host factors for SARS-CoV-2 entry in human lung and intestinal tissues, our results suggested that the dependency on furin cleavage of spike for efficient SARS-CoV-2 replication in the human lung was owing to a relatively low level of TMPRSS2 expression. Collectively, our study revealed critical host and viral determinants for efficient SARS-CoV-2 replication in the human lung,

which extended our understanding on the biology of SARS-CoV-2 infection and transmission.

HS is a linear polysaccharide that contains N-acetylglucosamine (GlcNAc) and glucuronic acid (GlcA), with groups of sulfate and other modifications[40]. It serves as a major adhesion receptor for a large number of viruses including human immunodeficiency virus, respiratory syncytial virus, and dengue virus[41–43]. In addition, HS also functions as an entry receptor for herpes simplex virus type 1[44]. In the context of human coronavirus, HS was previously reported to serve as an indispensable attachment receptor for HCoV-NL63[11]. By blocking the interaction between cell surface HS and virus spike, the attachment and infection of HCoV-NL63 was largely abolished[11]. Here, we demonstrated that blocking the interaction between cell surface HS and virus spike with exogenous HS pretreatment significantly reduced SARS-CoV-2 attachment on Calu3 cells by 48% ($p = 0.0012$). Meanwhile, removing cell surface HS by heparinase treatment reduced SARS-CoV-2 in the supernatant of Calu3 cells by 73% ($p = 0.0006$) at 24 hpi. These evidence suggested an essential role of HS in SARS-CoV-2 attachment and replication. As HS is ubiquitously expressed by most cell types, it may serve an important role during SARS-CoV-2 infection as ACE2 expression is relatively low in the lung and respiratory tract comparing to many other human tissues[18]. In this regard, blocking the interaction between cell surface HS and viral spike should be considered as a potential intervention strategy for COVID-19. Interestingly, as a common anticoagulant, heparin was applied in the treatment of COVID-19 coagulopathy and was found to associate with decreased mortality in severe cases[45]. In addition, heparin demonstrated potent antiviral activity against SARS-CoV-2 in vitro[46,47]. As heparin is structurally

similar with that of HS, its potential capacity on blocking SARS-CoV-2 infection should be further explored in in vivo or clinical models.

Sialic acid was a recognized host factor that served as an attachment factor or entry receptor for a number of coronaviruses including transmissible gastroenteritis virus, bovine coronavirus, HCoV-OC43, and HCoV-HKU1[13,14,48,49]. Recently, MERS-CoV spike was identified to have sialic acid-binding capacity and sialic acid depletion in Calu3 cells reduced MERS-CoV entry by >70% compared with mock-treatment[12]. However, the role of sialic acid in SARS-CoV-2 or SARS-CoV infection is unknown. Here, by using a combination of human lung epithelial cells, SLC35A1$^{KO}$ cells, and ex vivo human lung tissues, we demonstrated an unexpected role of sialic acid against SARS-CoV infection. In comparison with SARS-CoV, SARS-CoV-2 attachment and replication was less restricted by sialic acids. This might be explained by the higher ACE2-binding affinity by SARS-CoV-2 RBD comparing with SARS-CoV RBD[37,50]. Importantly, by utilizing ACE2 glycosylation mutants and SPR, we further demonstrated that the sialic acids present on ACE2 precluded perfect spike/ACE2-interaction during SARS-CoV-2 infection, potentially contributed by the N90 residue. Since sialic acids are abundantly expressed in the human respiratory tract, the capacity of SARS-CoV-2 to partly overcome sialic acid-mediated restriction may contribute to its efficient replication and person to person transmission in comparison to SARS-CoV. Interestingly, to promote virus replication, influenza viruses encode for viral neuraminidase for efficient virus release from the infected host cells. In addition, a number of bacteria also express sialidase[51]. In this regard, the modulatory effect of influenza viruses and bacteria on SARS-CoV-2 infection may deserve further investigations.

The "PRRA" amino-acid insertion at the $S_1/S_2$ junction of SARS-CoV-2 spike was recently identified and was widely speculated to facilitate virus replication or expand virus tropism[23,24,26]. In this study, we carefully evaluated the importance of the reported $S_1/S_2$ furin cleavage site in SARS-CoV-2 spike during SARS-CoV-2 infection. Interestingly, SARS-CoV-2 replicated better than SARS-CoV-2 $S_1/S_2$mut in the human lung epithelial cells (Calu3), which expressed low levels of TMPRSS2 and cathepsin L. At the same time, the furin inhibitor, dec-RVKR-cmk, reduced spike cleavage and limited SARS-CoV-2 replication in Calu3 but not VeroE6 cells. Our findings indicated that furin cleavage of the virus spike would be essential for efficient SARS-CoV-2 replication in cell lines without either a robustly operative plasma membrane or endosomal entry pathway. Previous studies suggested that human coronaviruses could utilize either the plasma membrane serine protease TMPRSS2 or endosomal cysteine proteases cathepsin L for spike activation and virus entry in cell lines[20–22]. However, TMPRSS2 played a more predominant role in virus replication and infection than cathepsin L in primary cells and animal models[52–54]. Here, our data showed that TMPRSS2 is expressed at a significantly higher level in the human intestinal tissues in comparison with the human lung tissues. In line with this finding, we further demonstrated that furin cleavage at $S_1/S_2$ junction of SARS-CoV-2 spike was required for efficient SARS-CoV-2 replication in the human lung but not intestine, potentially owing to the significantly lower TMPRSS2 expression in the lung comparing to the intestine. In this regard, the potential of TMPRSS2 and furin inhibition should be further explored as potential intervention strategies against SARS-CoV-2 since the respiratory tract is the primary target of the virus.

In summary, our study revealed essential host and viral determinants for SARS-CoV-2 infection of the human lung. These findings contributed to our understanding on the efficient replication of SARS-CoV-2 in the respiratory tract and suggested targets of intervention against COVID-19.

## Methods

**Viruses and biosafety**. SARS-CoV-2 HKU-001a (GenBank accession number MT230904) was isolated from the nasopharyngeal aspirate of a laboratory-confirmed COVID-19 patient in Hong Kong as previously described[32]. SARS-CoV GZ50 (GenBank accession number AY304495) was an archived clinical isolate at Department of Microbiology, HKU. MERS-CoV EMC/2012 was kindly provided by Dr. Ron Fouchier from Erasmus Medical Center. The influenza A virus strain A/Hong Kong/415742/2009(H1N1)pdm09 was isolated and archived at Department of Microbiology, HKU[55]. VeroE6 cells were used to culture SARS-CoV-2, SARS-CoV, and MERS-CoV. All viruses were titrated by plaque assays. SARS-CoV-2 HKU-001a $S_1/S_2$mut (GenBank accession number MT621560), which carried a 10 amino-acid deletion, was isolated by plaque purification from SARS-CoV-2 cultured in VeroE6 cells and subsequently sequenced (Supplementary Fig. 4). All experiments with infectious SARS-CoV-2, SARS-CoV-2 $S_1/S_2$mut, SARS-CoV, and MERS-CoV were performed following the approved standard operating procedures of the Biosafety Level 3 facility at Department of Microbiology, HKU.

**Cell cultures**. Calu3, Caco2, Huh7, BHK21, RK13, PK15, CRFK, 293 T, and VeroE6 were obtained from ATCC. The cells were maintained in minimum essential medium (MEM) (Gibco, MA, USA), Dulbecco's modified Eagle's medium (DMEM) (Gibco), or DMEM/F12 (Gibco) according to supplier's instructions. The cell lines used are routinely tested for mycoplasma and are maintained mycoplasma-free.

**Human ex vivo lung and intestine tissues and virus inoculation on tissues**. Human lung and intestine tissues in our ex vivo studies were acquired from patients undergoing surgeries at Queen Mary Hospital, Hong Kong as we previously described[31]. All donors provided written consent as approved by the Institutional Review Board of the University of Hong Kong/Hospital Authority Hong Kong West Cluster (UW13-364). The freshly obtained human lung or intestine tissues were immediately processed upon receipt into small rectangular pieces and were rinsed with advanced DMEM/F12 medium (Gibco) supplemented with 2 mM of 4-(2-hydroxyethyl)-1-piperazineethanesulfonic acid (HEPES) (Gibco), 1× GlutaMAX (Gibco), 100U/ml penicillin, and 100 μg/ml streptomycin. The processed specimens were infected with SARS-CoV, SARS-CoV-2, or SARS-CoV-2 $S_1/S_2$mut at an inoculum of $1 \times 10^7$PFU/ml per well. After infection, the inoculum was removed and the specimens were washed with phosphate-buffered saline (PBS). The infected human lung or intestine tissues were then cultured in advanced DMEM/F12 medium with 2 mM HEPES (Gibco), 1× GlutaMAX (Gibco), 100 U/ml penicillin, 100 μg/ml streptomycin, 20 μg/ml vancomycin, 20 μg/ml ciprofloxacin, 50 μg/ml amikacin, and 50 μg/ml nystatin under harvest.

**SARS-CoV-2-S-pseudoviruses and pseudovirus entry assays**. Pseudoviruses were generated in 293 T cells. In brief, 293 T cells were transfected with SARS-CoV-2-S, SARS-CoV-2-S($S_1/S_2$mut), or VSV-G plasmids with Lipofectamine 3000 (Thermo Fisher Scientific). At 24 h post transfection, the cells were infected with VSV-deltaG-firefly pseudotyped with VSV-G[56]. The cells were washed extensively after 2 h and fresh culture medium was added. The generated SARS-CoV-2-S-pseudoviruses, SARS-CoV-2-S($S_1/S_2$mut)-pseudoviruses, or VSV-G-pseudoviruses were harvested at 16 h post inoculation and were titrated with TCID$_{50}$. For pseudovirus entry assays, target cells were inoculated with pseudoviruses for 24 h, before washed and lysed for detection of luciferase signal with a luciferase assay system (E1501, Promega).

**Surface plasmon resonance**. SPR assays were carried out with a Biacore X100 system (GE Healthcare). In brief, SARS-CoV-2 RBD protein (SPD-C82E9, ACROBiosystems, DE, USA) was immobilized on a CM5 sensor chip (BR100012, GE Healthcare) to the target level of 1000 response unit (RU) via the amine coupling kit (BR-1000–50, GE Healthcare). For immobilization, carboxyl groups on the CM5 sensor chip surface were first activated by injection of N-hydroxysuccinimide and 1-ethyl-3-(3-dimethylaminopropyl)-carbodiimide hydrochloride in the amine coupling kit. Next, SARS-CoV-2 RBD protein at the concentration of 30 μg/ml in 10 mM sodium acetate buffer (pH 5.0) was flowed over the chip surface at a rate of 5 μl/min for 7 min, and to the target level of 1000 RU. Excess active ester groups on the sensor surface were capped by injection of 1 M ethanolamine (pH 8.5) at a flow rate of 5 μl/min for 7 min after unreacted protein was washed out. Binding experiments were performed at 25 °C in running buffer contained 10 mM HEPES pH7.2, 150 mM NaCl, and 0.05% Tween-20. Human ACE2 recombinant protein (10108-H08H-B, SinoBiological) were pretreated with NA from A. ureafaciens (P0722L, NEB) at 37 °C for 2 h before SPR assays. Serial dilutions of pretreated or control ACE2 were flowed through at concentrations ranging from 6.25 to 100 nM at a flow rate of 30 μl/min and real time RU was recorded. After each cycle, the sensor surface was regenerated by injection of 10 mM glycine-HCl (pH 2.0) for 2 × 30 seconds at a flow rate of

30 μl/min. The resulting data was fit to a 1:1 binding model using the Biacore X100 evaluation software (version 2.0.1, GE Healthcare).

**HS and heparinase treatment**. HS (HY-101916, MedChemExpress, NJ, USA) was incubated with viruses before infection at the indicated concentrations at room temperature for 1 h. For virus attachment assays, the pretreated viruses were incubated with cells at 0.2 MOI at 4 °C for 2 h, washed, and harvested. For replication assays, the pretreated viruses were incubated with cells at 0.2 MOI at 37 °C for 2 h. The cells were washed after 2 h and incubated in HS-supplemented culture media. Cell lysates and supernatants were harvested at 24 hpi. For heparinase assays, Calu3 or Caco2 cells were pretreated with Heparinase I (4U/ml) (P0735L, NEB, MA, USA), Heparinase III (0.5U/ml) (P0737L, NEB), or both at 37 °C for 1 h. Viruses were inoculated into the pretreated cells at 0.2 MOI for 2 h at 37 °C. Cell lysates and supernatants were harvested at 24 hpi. For HS treatment on ex vivo human lung tissues, SARS-CoV-2 and SARS-CoV at $1 \times 10^7$PFU/ml were pre-incubated with 1000 μg/ml HS for 1 h at room temperature. Lung tissues were inoculated with pretreated viruses for 2 h, washed, incubated in HS-supplemented culture media, and harvested at 24hpi.

**Neuraminidase treatment on cell lines and ex vivo human lung tissues**. To determine the role of sialic acids during SARS-CoV-2 and SARS-CoV infection in cell lines, Caco2 and Calu3 cells were pretreated with 12.5U NA from *A. ureafaciens* (P0722L, NEB) at 37 °C for 1 h before infection. The pretreated cells were challenged with SARS-CoV-2 or SARS-CoV at 0.2 MOI for 2 h at 4 °C for virus attachment assays. Alternatively, the pretreated cells were challenged with SARS-CoV-2 or SARS-CoV at 0.2 MOI for 2 h at 37 °C for virus replication assays. At 2hpi, the cells were washed and NA-supplemented culture media were added. Cell lysates and supernatants were collected at 24 hpi. For ex vivo human lung tissues, the lung specimens were pre-incubated with 50U NA for 1 h at 37 °C. Pretreated human lung tissues were challenged with SARS-CoV-2 and SARS-CoV at $1 \times 10^7$PFU/ml for 2 h. After virus inoculation, the lung tissues were washed, incubated in NA-supplemented culture media, and harvested at the indicated time points.

**Dec-RVKR-cmk treatment on Calu3 and VeroE6 cells**. Calu3 or VeroE6 cells were infected with SARS-CoV-2 for 1 h and were treated with dec-RVKR-cmk (3501, Tocris Bioscience, Bristol, United Kingdom) at 4 μM or 20 μM, or were mock-treated. At 24hpi, the cell lysates and supernatants were harvested for qRT-PCR quantification of virus replication. In parallel, cell lysates were harvested at 72hpi for western blot detection of spike cleavage.

**Sialic acid transporter SLC35A1 knockout cells**. SLC35A1 sgRNA oligos (5′-CACCGCCATAGCTTTAAGATACACA-3′ and 3′-CGGTATCGAAATTCTA TGTGTCAAA-5′) were annealed by heating to 95 °C for 5 minutes and cooling to 25 °C at 1.5 °C/minute. The lentiCRISPR v2 plasmid (Addgene) was digested by BsmBI (NEB) and purified by electrophoresis and gel purification. The digested lentiCRISPR v2 plasmid (Addgene) and annealed oligos were then ligated into SLC35A1-lentiCRISPR using the Quick Ligation Kit (M2200L, NEB). Ligation products were purified by the PureLink PCR purification kit (Thermo Fisher Scientific, MA, USA). Purified plasmids were transformed into DH5α competent cells and selected with ampicillin. At 14 hours after transformation, colonies were picked for sequence analysis using human U6 primer. Positive colonies were expanded and extracted using QIAGEN Plasmid Plus Kits (12963, QIAGEN, Hilden, Germany) to obtain the SLC35A1-lentiCRISPR plasmid. The SLC35A1-lentiCRISPR plasmid was co-transfected with packaging plasmids pMD2.G (Addgene) and psPAX2 (Addgene) into 293 T cells using Lipofectamine 3000 (L3000015, Thermo Fisher Scientific). The cell supernatant that contained SLC35A1-lentiCRISPR lentiviruses was harvested at 48 hpi and used for transduction. The transduced HEK293 cells were selected for 72 hours in 1 μg/ml puromycin and were limiting diluted in 96-well plates for clonal expansion. Depletion of SLC35A1 expression in these clones was determined by DNA sequencing. The phenotype of SLC35A1 knockout was confirmed by immunofluorescence staining for cell surface sialic acids with Sambucus Nigra (SNA) lectin (FL-1301, Vector Laboratories, CA, USA). In addition, the SLC35A1 knockout cells were functional evaluated by loss of capacity of supporting influenza virus H1N1 infection. For this purpose, SLC35A1 or control knockout cells were challenged with H1N1 at 0.1 MOI for 1 hour at 37 °C. At 24 hours post virus challenge, the cell lysates were harvested for qRT-PCR analysis of virus replication. To evaluate the role of sialic acid on SARS-CoV-2 or SARS-CoV attachment, SLC35A1- and control-knockout HEK293 cells were transfected with hACE2 plasmid or empty vector for 24 h. The transfected cells were challenged with SARS-CoV-2 or SARS-CoV at 0.2 MOI for 2 h at 4 °C. After 2 hours, the inoculum was removed and the cells were washed 3 times with PBS. Cell lysates were harvested in RLT buffer (Qiagen) for qRT-PCR analysis.

**RNA extraction and qRT-PCR**. RNA extraction and qRT-PCR were performed as we previously described[57,58]. Cell lysates and tissue samples were lysed by RLT buffer (Qiagen) and extracted with the RNeasy Mini kit (74106, Qiagen). Supernatant samples were lysed by AVL buffer (Qiagen) and extracted with the QIAamp Viral RNA Mini kit (52906, Qiagen). After RNA extraction, qRT-PCR was performed using the QuantiNova SYBR Green RT-PCR kit (208154, Qiagen) or the

QuantiNova Probe RT-PCR Kit (208354, Qiagen) with the LightCycler 480 Real-Time PCR System (Roche, Basel, Switzerland)[32,59]. All primer and probe sequences are provided in Supplementary Table 1.

**Plaque assay**. VeroE6 cells were seeded in 12-well plates 1 day before the experiment. The harvested supernatant samples were serially diluted and inoculated to the cells for 2 h at 37 °C. After inoculation, the cells were washed with PBS 3 times, and covered with 2% agarose/PBS mixed with 2× DMEM/2%FBS at 1:1 ratio. The cells were fixed after incubation at 37 °C for 72 h. Fixed samples were stained with 0.5% crystal violet in 25% ethanol/distilled water for 10 min for plaque visualization.

**Immunofluorescence staining**. Immunofluorescence staining was performed as we previously described with slight modifications[60]. In brief, infected cells were fixed overnight in 10% formalin. The fixed samples were washed with PBS and blocked by serum-free protein blocker (X0909, Dako, Glostrup, Denmark) and Sudan black B. The samples were then stained with an in-house rabbit anti-SARS-CoV-2-nucleocapsid (N) immune serum or an in-house rabbit anti-SARS-CoV-N immune serum for virus detection. The immune sera were generated by subcutaneously injecting purified SARS-CoV-2-N or SARS-CoV-N recombinant protein mixed with an equal volume of complete Freund's adjuvant (Sigma-Aldrich) into 4-6-week-old New Zealand white rabbit as we described in a previous study[32]. Sambucus Nigra lectin (SNA) from Vector Laboratories was used for sialic acid detection. Goat anti-rabbit IgG secondary antibody (A11034) was purchased from Thermo Fisher Scientific. The antifade mounting medium with 4′,6-Diami-dino-2-phenylindole dihydrochloride (DAPI, H-1200, Vector Laboratories) was used for mounting and DAPI staining. Images were taken with the Olympus BX53 fluorescence microscope (Olympus Life Science, Tokyo, Japan).

**Immunohistochemistry**. Immunohistochemistry was performed as we previously described with slight modifications[31]. Human lung tissue samples were collected and fixed overnight in 10% formalin. The fixed samples were embedded in paraffin by TP1020 Leica semi-enclosed benchtop tissue processor and sectioned with microtome (Thermo Fisher Scientific). Sectioned samples were prepared on glass slides and were dewaxed and dehydrated by serially diluted xylene, ethanol, and double-distilled water in sequence. Afterwards, the samples were co-heated together with antigen unmasking solution (H-3300, Vector Laboratories) at 85 °C for 90 s for antigen exposure, followed by blocking with 0.3% hydrogen peroxide for 30 min, and 1% BSA for 30 min. The in-house rabbit anti-SARS-CoV-2-N immune serum or in-house rabbit anti-SARS-CoV-N immune serum were applied as the primary antibodies and were incubated with the slides at 4 °C overnight. The signal was developed with the DAB (3,3'-diaminobenzidine) substrate kit (SK-4100, Vector Laboratories). Cell nuclei were labeled with Gill's haematoxylin. The slides were mounted with antifade mounting medium with DAPI (H-1200, Vector Laboratories). Images were taken with the Olympus BX53 light microscope (Olympus Life Science).

**Western blot**. Cells grown on 24-well plate were transfected with furin-flag and/or hACE2-V5 plasmids using Lipofectamine 3000 (Thermo Fisher Scientific). Cells were lysed by RIPA buffer (89901, Thermo Fisher Scientific) with protease inhibitor (4693169001, Roche, Basel, Switzerland) at 24 h post transfection. Proteins were separated with sodium dodecyl sulfate polyacrylamide gel electrophoresis (SDS-PAGE) and transferred to nitrocellulose membrane. Specific primary antibodies were incubated with the blocked membranes at 4 °C overnight, followed by horseradish peroxidase (HRP) conjugated secondary antibodies (Thermo Fisher Scientific) for 1 h at room temperature. The signal was developed by Immobilon Crescendo Western HRP Substrate (WBLUR0500, Merck Millipore, MA, USA) and detected using Alliance Q9 Advanced (Uvitec, Cambridge, UK). Rabbit anti-human furin antibody (ab183495, Abcam, Cambridge, UK) was used as the primary antibody for furin detection. Human ACE2 was detected with a rabbit anti-V5 antibody (MAB8926, R&D). Beta-actin was used as the loading control, which was detected by mouse anti-human beta-actin antibody (MAB8929, R&D). In the SARS-CoV-2 and SARS-CoV-2 $S_1/S_2$mut infection experiment in VeroE6, as well as the SARS-CoV-2 infection experiment in Calu3 and VeroE6 with or without dec-RVKR-cmk, the spike protein was detected with a rabbit anti-SARS-CoV-2 spike S2 antibody (40590-T62, Sino Biological).

**Flow cytometry**. BHK21 cells were transfected with ACE2 glycosylation mutants with Lipofectamine 3000 (Thermo Fisher Scientific). The cells were harvested at 24 hours post transfection. To determine the cell surface expression level of ACE2, the cells were detached with 10 mM ethylenediaminetetraacetic acid in PBS, fixed in 4% paraformaldehyde, followed by immunolabeling with an antibody against ACE2 (AF933, R&D Systems) without cell permeabilization. The flow cytometry was performed using a BD FACSCanto II flow cytometer (BD Biosciences) and data was analyzed using FlowJo X 10.0.7 (BD). The gating strategy was demonstrated in Supplementary Figure 3.

**Study approval**. All experiments in the study complied with the relevant ethical regulations for research. The use of human lung and intestine tissues for ex vivo studies was approved by the Institutional Review Board of the University of Hong Kong/Hospital Authority Hong Kong West Cluster (#UW 13-364).

**Statistical analysis**. Data represented mean and standard deviations. Statistical differences between two groups were evaluated with two-sided unpaired Student's $t$ test using GraphPad Prism 6. Statistical differences between three or more groups were evaluated with two-way or one-way analysis of variance using Graphpad Prism 6. Differences were considered statistically significant when $P < 0.05$.

**Reporting summary**. Further information on research design is available in the Nature Research Reporting Summary linked to this article.

## Data availability

The data that support the findings of this study are available from the corresponding authors upon reasonable request. Source data are provided with this paper.

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

## Acknowledgements

This study was partly supported by the donations of May Tam Mak Mei Yin, the Shaw Foundation of Hong Kong, Richard Yu and Carol Yu, Michael Seak-Kan Tong, Respiratory Viral Research Foundation Limited, Lee Wan Keung Charity Foundation Ltd, Hui Ming, Hui Hoy and Chow Sin Lan Charity Fund Limited, Chan Yin Chuen Memorial Charitable Foundation, Marina Man-Wai Lee, the Hong Kong Hainan Commercial Association South China Microbiology Research Fund, the Jessie & George Ho Charitable Foundation, Perfect Shape Medical Limited, Kai Chong Tong, Foo Oi Foundation Ltd, Tse Kam Ming Laurence, Betty Hing-Chu Lee, Ping Cham So, and Lo Ying Shek Chi Wai Foundation; and funding from the Consultancy Service for Enhancing Laboratory Surveillance of Emerging Infectious Diseases and Research Capability on Antimicrobial Resistance for Department of Health of the Hong Kong SAR Government; Health and Medical Research Fund (16150572); General Research Fund (17123920); the Theme-Based Research Scheme (T11/707/15) of the Research Grants Council; Hong Kong Special Administrative Region; Sanming Project of Medicine in Shenzhen, China (No. SZSM201911014); NIH R01AI139238, and the High Level-Hospital Program, Health Commission of Guangdong Province, China. The funding sources had no role in the study design, data collection, analysis, interpretation, or writing of the report.

## Author contributions

H.C., J.F.W.C., and K.Y.Y. designed the study. H.C., B.H., X.H., Y.C., Z.D., Y.W., H.S., D.Y., Y.H., X.Z., T.T.T.Y., J.P.C., A.J.Z., J.Z., S.Y., K.K.W.T., I.H.Y.C., K.Y.S., D.C.C.F., I.Y.H.W., A.T.L.N., T.T.C., S.Y.K.L., W.K.A., M.A.B., Z.C., K.H.K., J.F.W.C., and K.Y.Y. performed experiments, analyzed data, or provided key resources. H.C., J.F.W.C., and K.Y.Y. wrote the manuscript.

## Competing interests

The authors declare no competing interests.

## Additional information

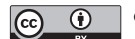

