## [Peer Review File · Nature Communications]

REVIEWER COMMENTS

Reviewer #1 (Remarks to the Author):

The manuscript by Chu and colleagues aims to investigate cellular factors required for SARS-CoV-2 attachment beyond the level of ACE2 (functional receptor). Here, the authors describe the role of heparin sulfate (HS) and sialic acid (SA) on the infectious entry and replication of SARS-CoV-2 and further demonstrate that the multibasic motif in the SARS-CoV-2 spike glycoprotein at the border between the S1 and the S2 subunit is required for efficient infection of ex vivo human lung but not intestinal cells. First, the authors show that pre-treatment of SARS-CoV-2 with soluble HS reduces infectivity. In addition, removal of cell surface HS by heparinases had the same effect. HS are used by many (corona) viruses as additional attachment factors and the involvement of HS in the entry of SARS-CoV-2 has been already speculated (e.g. Kim et al., 2020, Antiviral Res). Next, the authors investigated the role of SA and show that removal of cell surface SA by bacterial neuraminidase has no (Calu3) to moderate beneficial impact on SARS-CoV-2 infectivity. Far more interesting is the observation that infectivity of SARS-CoV can be robustly increased when SA are removed. Finally, the peculiar multibasic motif at the S1/S2 border of the SARS-CoV-2 spike protein was investigated. Here, the authors show that this motif is required for efficient infection of Calu3 and ex vivo human lung cells but not Vero E6, Caco2, Huh7 or ex vivo human intestinal cells

Altogether, the findings are of general interest but apart from the fact that cellular sialic acids seem to reduce SARS-CoV infectivity the findings are neither ground breaking (HS is used by many CoVs as an additional attachment factor, therefore involvement of HS in entry of SARS-CoV-2 is not surprising) nor novel (importance of multibasic motif for infection of human lung cells, as demonstrated by Kleine-Weber et al., 2020, Mol Cell). In my opinion this study would be a better fit for a more specialized journal.

Specific comments:

- Data presented in Figures 1 and 2 (and Figure 3 panel C) should also be shown as non-normalized data (e.g. genome copies per ml), as it has been done for Figures 3 (mainly), 4 and 5. This will help the reader to set the different viruses analyzed (SARS-CoV-2, SARS-CoV, MERS-CoV) into relation.

- The SARS-CoV-2 stocks have been prepared on Vero E6 cells (Line 363f). Did the authors sequence the viral spike gene, as it has been reported that the multibasic motif at the S1/S2 site can quickly mutate or even get lost upon propagation of the virus on Vero cells (Lau et al., 2020, Emerg Microbes and Infect). If not, this needs to be verified if, and if yes to which extent, the virus stock might contain viruses with a mutated S1/S2 site.

- The strong increase of SARS-CoV infectivity (and to a much lesser degree for SARS-CoV-2 in Caco2 cells) following removal of SA from the cell surface suggests that SA hinder efficient attachment of SARS-CoV to target cells. This observation is generally interesting and warrants further research. In their model presented in Figure 6, the authors suggest that SA of unrelated cell surface molecules restrain SARS-CoV/SARS-CoV-2 from efficient attachment to target cells. However, it is a more likely scenario that SA that are present on the cellular receptor ACE2 preclude perfect spike/ACE2-interaction. This could be investigated by quantitative assessment of S protein interaction with (i) ACE2 variants that are mutated in a way that they lack specific glycosylation sites and (ii) soluble ACE2 pre-treated with or without neuraminidase.

- The data presented in Figure 5B imply that only TMPRSS2 gene expression separates SARS-CoV-2-susceptible from non-susceptible cell lines, whereas gene expression levels for ACE2, cathepsin L, cathepsin B and furin are in the same range between the two groups. However, Vero E6 cells - which have not been tested but should have - do not express TMPRSS2 (Kleine-Weber et al, 2018, Sci Rep) and are still highly susceptible. Based on their data, the authors state that "These findings suggested that in model cell lines, furin cleavage at S1/S2 junction of spike was required for efficient SARS-CoV-2 replication in the absence of either a robust plasma membrane entry pathway or a robust endosomal entry pathway.", however S protein cleavage by furin has not been investigated. This should be done (Western blot).

- In addition to the previous point: The authors spend a lot of efforts in investigating the S1/S2 site without even looking at spike protein cleavage in the cell lines analyzed or to what extent wt and mutant "S1/S2mut" differ in their proteolytic processing.

- With respect to Figure 5A: For Calu3 cells mutant "S1/S2mut" infection is clearly reduced while for Caco2 and RK13 cells mutant "S1/S2mut" displays (slightly) better infectivity than SARS-CoV-2 containing wildtype spike protein. As a virus isolate (plaque purified) was used that also has additional mutations outside the spike gene, there is the possibility that they also modulate infectivity and thus preclude precise conclusions. Using reverse genetics to generate viruses that only differ in the cleavage site would have been the best way to investigate the S1/S2 site.

Reviewer #2 (Remarks to the Author):

Chu et al. report findings regarding the mechanistic implications of heparin sulfates, sialic acids, and furin sites in SARS-CoV-2 that should be of interest to the scientific community. The role of heparin sulfates was elucidated by the enzymatic “knock-down” of these sugars showing a decrease in infection. The authors demonstrate that unlike MERS-CoV, sialic sugars do not facilitate entry of the SARS-CoVs in a similar enzymatic “knock-down” assay. Finally, authors show that a furin-deficient SARS-CoV-2 replicate similarly in intestines but not in lung cells.

Requested revisions

1. Please include some discussion about the role of TMPRSS2 and how it may account for dependency on furin activation (and the subsequent lung vs intestine cell type replication-prone specificity).
2. Please cite the reference (<https://doi.org/10.1038/s41421-020-00192-8>) demonstrating how certain sulfated polysaccharides, including heparin, show potent anti-viral activity with no toxicity against SARS-CoV-2 in vitro.
3. Please cite the reference (<https://doi.org/10.1016/j.antiviral.2020.104873>) proposing heparin binding with the spike through both the RBD and furin cleavage site.

RE: [NCOMMS-20-25897A] Host and viral determinants for efficient SARS-CoV-2 infection of the human lung.

Response to reviewers' comments:

Reviewer #1 (Remarks to the Author):

The manuscript by Chu and colleagues aims to investigate cellular factors required for SARS-CoV-2 attachment beyond the level of ACE2 (functional receptor). Here, the authors describe the role of heparin sulfate (HS) and sialic acid (SA) on the infectious entry and replication of SARS-CoV-2 and further demonstrate that the multibasic motif in the SARS-CoV-2 spike glycoprotein at the border between the S1 and the S2 subunit is required for efficient infection of ex vivo human lung but not intestinal cells. First, the authors show that pre-treatment of SARS-CoV-2 with soluble HS reduces infectivity. In addition, removal of cell surface HS by heparinases had the same effect. HS are used by many (corona) viruses as additional attachment factors and the involvement of HS in the entry of SARS-CoV-2 has been already speculated (e.g. Kim et al., 2020, Antiviral Res). Next, the authors investigated the role of SA and show that removal of cell surface SA by bacterial neuraminidase has no (Calu3) to moderate beneficial impact on SARS-CoV-2 infectivity. Far more interesting is the observation that infectivity of SARS-CoV can be robustly increased when SA are removed. Finally, the peculiar multibasic motif at the S1/S2 border of the SARS-CoV-2 spike protein was investigated. Here, the authors show that this motif is required for efficient infection of Calu3 and ex vivo human lung cells but not Vero E6, Caco2, Huh7 or ex vivo human intestinal cells. Altogether, the findings are of general interest but apart from the fact that cellular sialic acids seem to reduce SARS-CoV infectivity the findings are neither ground breaking (HS is used by many CoVs as an additional attachment factor, therefore involvement of HS in entry of SARS-CoV-2 is not surprising) nor novel (importance of multibasic motif for infection of human lung cells, as demonstrated by Kleine-Weber et al., 2020, Mol Cell). In my opinion this study would be a better fit for a more specialized journal.

Specific comments:

- Data presented in Figures 1 and 2 (and Figure 3 panel C) should also be shown as non-normalized data (e.g. genome copies per ml), as it has been done for Figures 3 (mainly), 4 and 5. This will help the reader to set the different viruses analyzed (SARS-CoV-2, SARS-CoV, MERS-CoV) into relation.
Response: Thank you for the useful comment. We have revised the figures. Results in Figure 1, Figure 2, and Figure 3c are now shown as virus gene copy (for cell lysate samples) or virus gene copy per mL (for supernatant samples).

- The SARS-CoV-2 stocks have been prepared on Vero E6 cells (Line 363f). Did the authors sequence the viral spike gene, as it has been reported that the multibasic motif at the S1/S2 site can quickly mutate or even get lost upon propagation of the virus on Vero cells (Lau et al., 2020, Emerg Microbes and Infect). If not, this needs to be verified if, and if yes to which extent, the virus stock might contain viruses with a mutated S1/S2 site.

Response: Thank you for the suggestion. We are fully aware of this potential issue. After plaque purification, both WT and S1/S2 viruses were whole-genome sequenced using an Oxford Nanopore MinION device supplemented by Sanger sequencing according to our established protocols^{1,2}. After preparing the stocks from the purified viruses, the stock viruses were sequenced with Sanger sequencing for the region spanning the S1/S2 mutation site (Sanger sequencing primer: Forward-5'-CAGGCTGTTTAATAGGGGC-3', Reverse-5'-CTGATGTCTTGGTCATAGACAC-3') to confirm the identity of the viruses.

In addition to sequencing, we have designed qPCR primers with the reverse primer located on the S1/S2 mutation site (Forward-5'-CAGGCTGTTTAATAGGGGC-3', Reverse-5'-CTACTACTACGTGCCCCGAGGAGA-3'). For the primary stocks, we calculated the ratio between the infectious titer determined by plaque assays and the genome copy determined by qPCR. Our result showed that for 1×10^7 PFU/ml of WT virus, qPCR resulted in approximately 5×10^9 copy/ml

of genome copies. In contrast, for 1×10^7 PFU/ml of S1/S2 mutant virus, qPCR resulted in no values or values below those of the no template controls.

We further modified our routine coronavirus culture protocol for generating SARS-CoV-2. For MERS-CoV and SARS-CoV, we inoculate VeroE6 cells at 0.05 MOI and harvest the supernatant at 72 hours post inoculation. For SARS-CoV-2, we inoculate VeroE6 at 0.5 MOI and harvest the viruses at 20 hours post inoculation by breaking the cells with free-thaw cycles. With our modified protocol, the ratio between the infectious titer and the genome copy (using the above-mentioned primer) of the WT virus remained stable between the primary stock and the subsequent passage.

With these precautions, we have made sure that our stock WT viruses remained the WT for the duration of the study.

- The strong increase of SARS-CoV infectivity (and to a much lesser degree for SARS-CoV-2 in Caco2 cells) following removal of SA from the cell surface suggests that SA hinder efficient attachment of SARS-CoV to target cells. This observation is generally interesting and warrants further research. In their model presented in Figure 6, the authors suggest that SA of unrelated cell surface molecules retrain SARS-CoV/SARS-CoV-2 from efficient attachment to target cells. However, it is a more likely scenario that SA that are present on the cellular receptor ACE2 preclude perfect spike/ACE2-interaction. This could be investigated by quantitative assessment of S protein interaction with (i) ACE2 variants that are mutated in a way that they lack specific glycosylation sites and (ii) soluble ACE2 pre-treated with or without neuraminidase.

Response: We agree with the reviewer that it is important to identify whether sialic acid on unrelated cell surface molecules or sialic acid on ACE2 restrain SARS-CoV-2/SARS-CoV from their efficient attachment to target cells.

As recommended by the reviewer, we investigated this question by (i) evaluating SARS-CoV-2 entry/infection in cells expressing ACE2 mutants that lack specific glycosylation sites and (ii) evaluating the binding between ACE2 with or without neuraminidase pre-treatment to SARS-CoV-2 RBD.

(i) First, we evaluated SARS-CoV-2 entry/infection in cells expressing ACE2 mutants that lack specific glycosylation sites. We synthesized ACE2 mutants according to recently published literature that identified ACE2 glycosylation sites³⁻⁵.

Specifically, N53Q, N90Q, N103Q, N322Q, N432Q, N546Q, and N690Q ACE2 mutants were individually generated to obtain N-linked glycosylation ACE2 mutants. The N (Asparagine) to Q (Glutamine) mutation was performed individually at each of the seven sites to remove the glycosylation motif, which was suggested by previous studies^{6,7}. Two of these sites, N90 and N322, were predicted to form interactions with the spike protein. Inter-molecular glycan-glycan interactions were predicted between the glycans at N546 of ACE2 and those at N74 and N165 of spike protein. In addition to the N-linked glycosylation sites, O-linked glycosylation sites were identified at S155, T496, and T730³⁻⁵. We generated S155A, T496A, and T730A mutations to remove these O-linked glycosylation motifs as suggested by previous studies^{8,9}. Moreover, we generated “del N”, “del O”, and “del N+O” mutants that simultaneously removed all N-linked, all O-linked, and all N- and O-linked glycosylation sites, respectively.

We evaluated these ACE2 variants on their capacity to support SARS-CoV-2 entry and SARS-CoV-2 replication.

To evaluate SARS-CoV-2 entry, we developed and constructed VSV-based SARS-CoV-2-S-pseudoviruses. We transfected these ACE2 variants to BHK21 cells and inoculated the cells with SARS-CoV-2-S-pseudoviruses at 24 hours post transfection. Pseudovirus entry was determined based on the level of luciferase signal detected at 24 hours post inoculation. Our results revealed a number of important findings. First, SARS-CoV-2-S-pseudovirus entry was significantly more efficient in

cells transfected with ACE2 variants N90Q and N546Q, which increased virus entry by 60% ($p=0.0189$) and 68% ($p=0.006$) for N90Q and N546Q, respectively (Fig. 4c). Second, none of the single mutations significantly reduced SARS-CoV-2-S-pseudovirus entry. This suggested that individually removing any of the predicted glycosylation sites did not significantly reduce SARS-CoV-2 entry. Third, while removing all N-linked (del N) or O-linked (del O) glycosylation sites did not impact SARS-CoV-2-S-pseudovirus entry, simultaneously removing all N- and all O-linked glycosylation sites dramatically reduced SARS-CoV-2-S-pseudovirus entry ($p<0.0001$).

To evaluate SARS-CoV-2 replication, we transfected the ACE2 variants to BHK21 cells and inoculated the cells with SARS-CoV-2. Samples were harvested at 24 hours post infection and virus replication in the cell lysates and supernatants were quantified with qPCR. The results were largely in line with those obtained in the pseudovirus entry assays. In particular, simultaneously removing all N- and all O-linked glycosylation sites dramatically reduced SARS-CoV-2 genome copy in both cell lysate and supernatant samples. In this setting, the N90Q but not the N546 mutant demonstrated a statistically significant increase over the WT control in the cell lysate samples (Fig. 4e).

Overall, our results by using the ACE2 glycosylation variants indicated that sialic acids at the N90 position of ACE2 have the most significant role in precluding a perfect ACE2-spike interaction during SARS-CoV-2 infection.

(ii) By introducing the N to Q or S/T to A mutations at the predicted ACE2 glycosylation sites, all glycan modifications in addition to the sialic acid groups are abolished at the modified site. To directly address the role of sialic acids present on ACE2 in spike/ACE2-interaction, we performed surface plasmon resonance (SPR) to determine the protein-protein binding affinity between SARS-CoV-2 spike receptor binding domain (RBD) and ACE2 with or without pretreating ACE2 with neuraminidase.

On our hand, human ACE2 binds to SARS-CoV-2 RBD with a dissociation constant (K_d) of 37.4nM, which is consistent with other SPR studies^{10,11} (Fig. 4g). Importantly, after neuraminidase pretreatment, human ACE2 binds to SARS-CoV-2 RBD at a higher binding affinity ($K_d=16.2nM$) (Fig. 4h). This finding clearly indicates that the sialic acids present on the cellular receptor ACE2 precluded perfect spike/ACE2-interaction.

- The data presented in Figure 5B imply that only TMPRSS2 gene expression separates SARS-CoV-2-susceptible from non-susceptible cell lines, whereas gene expression levels for ACE2, cathepsin L, cathepsin B and furin are in the same range between the two groups. However, Vero E6 cells - which have not been tested but should have - do not express TMPRSS2 (Kleine-Weber et al, 2018, Sci Rep) and are still highly susceptible. Based on their data, the authors state that "These findings suggested that in model cell lines, furin cleavage at S1/S2 junction of spike was required for efficient SARS-CoV-2 replication in the absence of either a robust plasma membrane entry pathway or a robust endosomal entry pathway.", however S protein cleavage by furin has not been investigated. This should be done (Western blot).

Response: Thank you for the suggestion. We do not intend to address SARS-CoV-2 permissiveness in this study and we do not intend to claim that only TMPRSS2 gene expression separates SARS-CoV-2-susceptible from non-susceptible cell lines because we are fully aware of the high susceptibility of VeroE6 cells to the virus. After reviewing the presented data in Fig. 5b, we agree with the reviewer that the currently presented data can mislead the audience to the above-mentioned message. We have now removed the results for the non-permissive cells from Fig. 5b since they are irrelevant to the theme of Fig. 5. In addition, as suggested by the reviewer, we measured the expression of different proteases in VeroE6. The results demonstrated an undetectable level of TMPRSS2 but very high level of cathepsin L expression in VeroE6. The revised and added data are now presented as Fig. 6 of the revised manuscript.

As suggested by the reviewer, we further investigated the role of furin cleavage in a new experiment.

We examined the replication and spike cleavage of SARS-CoV-2 in Calu3 cells with or without the presence of the furin inhibitor, dec-RVKR-cmk. Our results demonstrated that dec-RVKR-cmk reduced S cleavage and inhibited SARS-CoV-2 replication in Calu3 cells in a dose-dependent manner (Fig. S5a,b). In contrast, dec-RVKR-cmk did not inhibit spike cleavage or SARS-CoV-2 replication in VeroE6 cells (Fig. S5c,d), which is in keeping with our hypothesis since VeroE6 robustly expressed cathepsin L.

Collectively, by following the suggestions from the reviewer, the revised and the added data can now better demonstrate the role of furin during SARS-CoV-2 infection.

- In addition to the previous point: The authors spend a lot of efforts in investigating the S1/S2 site without even looking at spike protein cleavage in the cell lines analyzed or to what extent wt and mutant "S1/S2mut" differ in their proteolytic processing.

Response: Thank you for the useful suggestion. Accordingly, we performed Western blots on SARS-CoV-2-WT and SARS-CoV-2-S1/S2mut in infected VeroE6 cells. Our results demonstrated a clear difference on spike cleavage between SARS-CoV-2-WT and SARS-CoV-2-S1/S2mut. The lack of cleavage at the S1/S2 site for SARS-CoV-2-S1/S2mut is consistent with other S1/S2 mutants, which are well described in the literature¹²⁻¹⁶. The result is now included as Fig. S3 of the revised manuscript.

- With respect to Figure 5A: For Calu3 cells mutant "S1/S2mut" infection is clearly reduced while for Caco2 and RK13 cells mutant "S1/S2mut" displays (slightly) better infectivity than SARS-CoV-2 containing wildtype spike protein. As a virus isolate (plaque purified) was used that also has additional mutations outside the spike gene, there is the possibility that they also modulate infectivity and thus preclude precise conclusions. Using reverse genetics to generate viruses that only differ in the cleavage site would have been the best way to investigate the S1/S2 site.

Response: We understand the concern of the reviewer and we agree that reverse genetics would have been the best way to investigate this question.

Nevertheless, a number of evidence supported our findings in Figure 5.

First, we performed additional entry assays using pseudoviruses with SARS-CoV-2-WT spike or SARS-CoV-2-S1/S2 mutant spike. We evaluated pseudovirus entry in Caco2, Calu3, RK13, and VeroE6 cells since these are the cell types that demonstrated the largest differences between WT and S1/S2mutant virus in Fig. 5a. Our results demonstrated that the S1/S2mutant pseudovirus entered cells less efficiently in comparison to the WT pseudovirus in Calu3 cells. In contrast, the S1/S2mutant pseudovirus entered cells more efficiently in comparison to the WT pseudovirus in VeroE6 and RK13 cells. In addition, the S1/S2mutant pseudovirus has a trend of more efficient cell entry in Caco2 cells in comparison to the WT pseudovirus although the difference did not reach statistical significance. Overall, the trend of pseudovirus entry with SARS-CoV-2-WT spike or SARS-CoV-2-S1/S2 mutant spike was in agreement with our presented data in Fig. 5a. The results are now included as Fig. S4 of the revised manuscript.

Second, the three nonsynonymous changes as indicated in Fig. S2 have not been reported to affect virus replication. In addition, it is unlikely that these changes will result in a cell-type dependent effect.

Third, a recent preprint¹⁵ posted on August 26th (our study was submitted on June 29th and was online on Research Square on July 13th) was able to recapitulate our findings from VeroE6 and Calu3 using recombinant SARS-CoV-2 with mutations at the S1/S2 cleavage site. Importantly, our study further provided data from SARS-CoV-2-infected *ex vivo* human lung and intestinal tissues, which can better represent the scenario in the lungs and intestines of human COVID-19 patients.

Reviewer #2 (Remarks to the Author):

Chu et al. report findings regarding the mechanistic implications of heparin sulfates, sialic acids, and furin sites in SARS-CoV-2 that should be of interest to the scientific community. The role of heparin sulfates was elucidated by the enzymatic “knock-down” of these sugars showing a decrease in infection. The authors demonstrate that unlike MERS-CoV, sialic sugars do not facilitate entry of the SARS-CoVs in a similar enzymatic “knock-down” assay. Finally, authors show that a furin-deficient SARS-CoV-2 replicate similarly in intestines but not in lung cells.

Requested revisions

1. Please include some discussion about the role of TMPRSS2 and how it may account for dependency on furin activation (and the subsequent lung vs intestine cell type replication-prone specificity).

Response: Thank you for the suggestion. We have revised our discussion on the role of TMPRSS2 and its relationship with the dependency on furin activation.

Line 505-520 (of the text with track change): *“Interestingly, SARS-CoV-2 replicated better than SARS-CoV-2 S₁/S₂mut in the human lung epithelial cells (Calu3), which expressed low levels of TMPRSS2 and cathepsin L. At the same time, the furin inhibitor, dec-RVCR-cmk, reduced spike cleavage and limited SARS-CoV-2 replication in Calu3 but not VeroE6 cells. These findings indicated that furin cleavage of the virus spike would be essential for efficient SARS-CoV-2 replication in cell lines without either a robustly operative plasma membrane or endosomal entry pathway. Previous studies suggested that human coronaviruses could utilize either the plasma membrane serine protease TMPRSS2 or endosomal cysteine proteases cathepsin L for spike activation and virus entry in cell lines¹⁷⁻¹⁹. However, TMPRSS2 played a more predominant role in virus replication and infection than cathepsin L in primary cells and animal models²⁰⁻²². Here, our data showed that TMPRSS2 is expressed at a significantly higher level in the human intestinal tissues in comparison to the human lung tissues. In line with this finding, we further demonstrated that furin cleavage at S₁/S₂ junction of SARS-CoV-2 spike was required for efficient SARS-CoV-2 replication in the human lung but not intestine, potentially due to the significantly lower TMPRSS2 expression in the lung comparing to the intestine.”*

2. Please cite the reference (<https://doi.org/10.1038/s41421-020-00192-8>) demonstrating how certain sulfated polysaccharides, including heparin, show potent anti-viral activity with no toxicity against SARS-CoV-2 in vitro.

Response: Thank you for the suggestion. We have cited the suggested reference in the text.

3. Please cite the reference (<https://doi.org/10.1016/j.antiviral.2020.104873>) proposing heparin binding with the spike through both the RBD and furin cleavage site.

Response: Thank you for the suggestion. We have cited the suggested reference in the text.

References:

- 1 Chan, J. F. *et al.* A familial cluster of pneumonia associated with the 2019 novel coronavirus indicating person-to-person transmission: a study of a family cluster. *Lancet*, doi:10.1016/S0140-6736(20)30154-9 (2020).
- 2 Chu, H. *et al.* Comparative tropism, replication kinetics, and cell damage profiling of SARS-CoV-2 and SARS-CoV with implications for clinical manifestations, transmissibility, and laboratory studies of COVID-19: an observational study. *Lancet Microbe* **1**, e14-e23, doi:10.1016/S2666-5247(20)30004-5 (2020).
- 3 Asif Shajahan *et al.* Comprehensive characterization of N- and O- glycosylation of SARS-CoV-2 human receptor angiotensin converting enzyme 2. *bioRxiv*, doi:<https://doi.org/10.1101/2020.05.01.071688> (2020).
- 4 Zhao, P. *et al.* Virus-Receptor Interactions of Glycosylated SARS-CoV-2 Spike and Human ACE2 Receptor. *Cell Host Microbe*, doi:10.1016/j.chom.2020.08.004 (2020).
- 5 Zhao, P. *et al.* Virus-Receptor Interactions of Glycosylated SARS-CoV-2 Spike and Human ACE2 Receptor. *bioRxiv*, doi:10.1101/2020.06.25.172403 (2020).
- 6 Graff, J. *et al.* Mutations within potential glycosylation sites in the capsid protein of hepatitis E virus prevent the formation of infectious virus particles. *J Virol* **82**, 1185-1194, doi:10.1128/JVI.01219-07 (2008).
- 7 Panda, A., Elankumaran, S., Krishnamurthy, S., Huang, Z. & Samal, S. K. Loss of N-linked glycosylation from the hemagglutinin-neuraminidase protein alters virulence of Newcastle disease virus. *J Virol* **78**, 4965-4975, doi:10.1128/jvi.78.10.4965-4975.2004 (2004).
- 8 Sadeghi, H. & Birnbaumer, M. O-Glycosylation of the V2 vasopressin receptor. *Glycobiology* **9**, 731-737, doi:10.1093/glycob/9.7.731 (1999).
- 9 Zhang, Y., Liu, R., Ni, M., Gill, P. & Lee, A. S. Cell surface relocalization of the endoplasmic reticulum chaperone and unfolded protein response regulator GRP78/BiP. *J Biol Chem* **285**, 15065-15075, doi:10.1074/jbc.M109.087445 (2010).
- 10 Shang, J. *et al.* Structural basis of receptor recognition by SARS-CoV-2. *Nature* **581**, 221-224, doi:10.1038/s41586-020-2179-y (2020).
- 11 Wrapp, D. *et al.* Cryo-EM structure of the 2019-nCoV spike in the prefusion conformation. *Science* **367**, 1260-1263, doi:10.1126/science.abb2507 (2020).
- 12 Hoffmann, M., Kleine-Weber, H. & Pohlmann, S. A Multibasic Cleavage Site in the Spike Protein of SARS-CoV-2 Is Essential for Infection of Human Lung Cells. *Mol Cell* **78**, 779-784 e775, doi:10.1016/j.molcel.2020.04.022 (2020).
- 13 Walls, A. C. *et al.* Structure, Function, and Antigenicity of the SARS-CoV-2 Spike Glycoprotein. *Cell* **181**, 281-292 e286, doi:10.1016/j.cell.2020.02.058 (2020).
- 14 Zhu Y *et al.* The S1/S2 boundary of SARS-CoV-2 spike protein modulates cell entry pathways and transmission. *bioRxiv*, doi:doi.org/10.1101/2020.08.25.266775 (2020).
- 15 Johnson, B. A. *et al.* Furin Cleavage Site Is Key to SARS-CoV-2 Pathogenesis. *bioRxiv*, doi:10.1101/2020.08.26.268854 (2020).
- 16 Xia, S. *et al.* The role of furin cleavage site in SARS-CoV-2 spike protein-mediated membrane fusion in the presence or absence of trypsin. *Signal Transduct Target Ther* **5**, 92, doi:10.1038/s41392-020-0184-0 (2020).

- 17 Simmons, G. *et al.* Inhibitors of cathepsin L prevent severe acute respiratory syndrome coronavirus entry. *Proc Natl Acad Sci U S A* **102**, 11876-11881, doi:10.1073/pnas.0505577102 (2005).
- 18 Shulla, A. *et al.* A transmembrane serine protease is linked to the severe acute respiratory syndrome coronavirus receptor and activates virus entry. *J Virol* **85**, 873-882, doi:10.1128/JVI.02062-10 (2011).
- 19 Shirato, K., Kawase, M. & Matsuyama, S. Middle East respiratory syndrome coronavirus infection mediated by the transmembrane serine protease TMPRSS2. *J Virol* **87**, 12552-12561, doi:10.1128/JVI.01890-13 (2013).
- 20 Zhou, Y. *et al.* Protease inhibitors targeting coronavirus and filovirus entry. *Antiviral Res* **116**, 76-84, doi:10.1016/j.antiviral.2015.01.011 (2015).
- 21 Simmons, G. *et al.* Different host cell proteases activate the SARS-coronavirus spike-protein for cell-cell and virus-cell fusion. *Virology* **413**, 265-274, doi:10.1016/j.virol.2011.02.020 (2011).
- 22 Iwata-Yoshikawa, N. *et al.* TMPRSS2 Contributes to Virus Spread and Immunopathology in the Airways of Murine Models after Coronavirus Infection. *J Virol* **93**, doi:10.1128/JVI.01815-18 (2019).

REVIEWER COMMENTS

Reviewer #1 (Remarks to the Author):

The authors have addressed all my points and added additional data, which significantly improved the manuscript. I have only one minor point with respect to the ACE2 mutants. Based on the data, ACE2 mutant del N+O is the only construct that shows reduced potential to serve as SARS-CoV-2 receptor. Although the Western blot data suggest that all ACE2 constructs are expressed at comparable levels in cell lysates it is not clear whether all variants are efficiently transported to the plasma membrane. It might be possible that surface expression of mutant del N+O is reduced and that this is the reason for decreased entry of pseudotypes bearing SARS2-S (and not reduced interaction). This question can be easily addressed using flow cytometry and such information would strengthen the conclusions drawn from the pseudotype data.

If the authors can address this point, I can recommend the manuscript for publication.

RE: [NCOMMS-20-25897B] Host and viral determinants for efficient SARS-CoV-2 infection of the human lung.

Response to reviewers' comments:

REVIEWER COMMENTS

Reviewer #1 (Remarks to the Author):

The authors have addressed all my points and added additional data, which significantly improved the manuscript. I have only one minor point with respect to the ACE2 mutants. Based on the data, ACE2 mutant del N+O is the only construct that shows reduced potential to serve as SARS-CoV-2 receptor. Although the Western blot data suggest that all ACE2 constructs are expressed at comparable levels in cell lysates it is not clear whether all variants are efficiently transported to the plasma membrane. It might be possible that surface expression of mutant del N+O is reduced and that this is the reason for decreased entry of pseudotypes bearing SARS2-S (and not reduced interaction). This question can be easily addressed using flow cytometry and such information would strengthen the conclusions drawn from the pseudotype data.

If the authors can address this point, I can recommend the manuscript for publication.

Response: Thank you for the useful suggestion. We transfected the ACE2 glycosylation mutants into BHK21 cells. At 24 hours post transfection, we harvested the cells and measured cell surface expression of ACE2 with flow cytometry. Our results demonstrated that the cell surface expression of N690Q ($p=0.0452$), S155A ($p=0.0042$), T496A ($p=0.0143$), del N ($p<0.0001$), del O ($p<0.0001$), and del N+O ($p<0.0001$) ACE2 mutants were lower than that of the WT ACE2. Among them, the del N+O ACE2 mutant had the lowest cell surface expression among all ACE2 mutants tested, which was 18.8% of the WT ACE2. The low cell surface expression of the del N+O ACE2 mutant may explain the reduced entry of SARS-CoV-2-S pseudovirus in del N+O ACE2 mutant-transfected BHK21 cells. This new result is now presented as Figure S2 of the revised manuscript.

REVIEWERS' COMMENTS

Reviewer #1 (Remarks to the Author):

The authors have addressed my remaining point and analyzed cell surface expression of the ACE2 mutants. The new data indicate that mutant "del N+O", which is less efficiently engaged by SARS-2-S, shows decreased cell surface expression. The new data are important for data interpretation and are discussed by the authors. I appreciate the extra efforts made by the authors and can recommend the manuscript for publication.

RE: [NCOMMS-20-25897C] Host and viral determinants for efficient SARS-CoV-2 infection of the human lung.

Response to reviewers' comments:

Reviewer #1 (Remarks to the Author):

The authors have addressed my remaining point and analyzed cell surface expression of the ACE2 mutants. The new data indicate that mutant "del N+O", which is less efficiently engaged by SARS-2-S, shows decreased cell surface expression. The new data are important for data interpretation and are discussed by the authors. I appreciate the extra efforts made by the authors and can recommend the manuscript for publication.

Response: We thank the reviewer for the comments provided along the review process. The suggested experiments were constructive and have substantially improved the quality of the manuscript.